# I wanna hold your hand: Handholding is preferred over gentle stroking for emotion regulation

Haran Sened[1]*, Chen Levin[1], Manar Shehab[1], Naomi Eisenberger[2], Simone Shamay-Tsoory[1]

1 Department of Psychology, University of Haifa, Haifa, Israel, 2 Department of Psychology, University of California, Los Angeles, California, United States of America

* haranse@gmail.com

## Abstract

Social touch is an important form of interpersonal emotion regulation. In recent years, the emotion regulation effects of two types of touch have been studied extensively: handholding and stroking (specifically of skin with C-tactile afferents on the forearm, i.e. C-touch). While some studies compare their effectiveness, with mixed results, no study to date has examined which type of touch is subjectively preferred. Given the potential bidirectional communication provided by handholding, we hypothesized that to regulate intense emotions, participants would prefer handholding. In four pre-registered online studies (total N = 287), participants rated handholding and stroking, presented in short videos, as emotion regulation methods. Study 1 examined touch reception preference in hypothetical situations. Study 2 replicated Study 1 while also examining touch *provision* preferences. Study 3 examined touch reception preferences of participants with blood/injection phobia in hypothetical injection situations. Study 4 examined types of touch participants who have recently given birth recalled receiving during childbirth and their hypothetical preferences. In all studies, participants preferred handholding over stroking; participants who have recently given birth reported receiving handholding more than stroking. This was especially evident in Studies 1–3 in emotionally intense situations. These results demonstrate that handholding is preferred over stroking as a form of emotion regulation, especially in intense situations, and support the importance of two-way sensory communication for emotion regulation via touch. We discuss the results and possible additional mechanisms, including top-down processing and cultural priming.

## Introduction

Touch is an important form of interaction in humans as well as in other species [1]. Interpersonal touch serves a variety of functions, including emotion communication [2] and modulating interpersonal bonds [3]. One major reason humans touch one another is to help regulate subjective experiences such as negative emotions and pain, sometimes referred to as consoling or comforting touch. Importantly, people can attempt to use different types of touch to regulate

**Data Availability Statement:** All four studies were preregistered on the Open Science Framework: Study 1: https://osf.io/x95f4/?view_only= e272eb9ccc2a46668114685dcb8988e5 Study 2:

https://osf.io/nzg6v/?view_only=
b9cf1e802d894ce387c80c47ffd7bc97 Study 3:
https://osf.io/8n3pu/?view_only=
f23a2e3d485946ffb95d32c02e469b85 Study 4:
https://osf.io/74szg/?view_only=
4b12d6d2a8a04483abbd6ea0c8c225ca. Study
data, analysis code and touch videos for Studies 1-
3 are available at the link below, except for touch
videos for Study 4 which were not posted as
participants who filmed the videos did not consent
to wide distribution. https://osf.io/nju48/?view_
only=81dadc972f6545f19d107c83d8dba368.

**Funding:** The work was funded by a MINDSS grant
awarded to the first author by the University of
Haifa and by the U.S. - Israel Binational Science
Fund (BSF) grant 2015068 awarded to the fourth
and fifth authors. The funders had no role in study
design, data collection and analysis, decision to
publish, or preparation of the manuscript.

**Competing interests:** The authors have declared
that no competing interests exist.

each other's emotions; each type of touch might be performed on different body locations, in different social contexts, and may be associated with different emotional experiences [4].

Fotopoulou and colleagues [5] argue that one of the functions of touch is homeostatic regulation. In homeostatic regulation, touch helps reset biological functioning to a fixed setpoint after an acute disturbance in homeostasis, as in warming up a cold baby. Shamay-Tsoory and Eisenberger [6] suggest that this kind of regulation involves a feedback loop: one person experiences an acute, unwanted experience (e.g. pain, or a negative emotion) while another person senses that and is motivated to provide comforting touch. The first person, in turn, perceives this touch, which serves to diminish emotional intensity. This feedback loop is homeostatic as it includes a corrective mechanism: feedback from a distressed target allows the toucher to adapt the touch accordingly. Thus, for homeostatic regulation to occur, both parties must establish two-way communication. This suggests that forms of touch which allow for better two-way feedback would be preferable over others, when trying to regulate intense emotional experiences and pain.

Two types of touch have been thoroughly examined in recent years using traditional paradigms testing the effectiveness of touch in distress regulation: handholding and gentle stroking.

Handholding entails grasping the other person's hand in varying degrees of strength. Studies have found that handholding can reduce pain [7], reduce pre-surgical anxiety [8], and lessen the emotional pain of recalling negative emotional experiences [9]. Handholding has been shown to attenuate pain-related activation in the posterior insula, the anterior cingulate cortex, the orbitofrontal cortex and the prefrontal cortex [10].

Gentle stroking entails slowly stroking skin regions with C-tactile afferents [11], such as hairy skin on the forearm. Although slow stroking activates all types of mechanoreceptive afferents, it is also referred to as C-touch (as C fibers respond optimally to this type of touch). At normal skin temperature (but not at very high or low temperatures), CT afferent firing is correlated with touch pleasantness [12]. Other studies have found that this kind of stroking, activates regions in the posterior insular cortex and the mid-anterior orbitofrontal cortex more than other types of touch [13,14]. Moreover, stroking activates social reward mechanisms [15], is associated with reduced feelings of social exclusion [16], although some of its effects may be attenuated by attachment anxiety[17,18]. It has also been suggested that stroking may be associated with elevated oxytocin levels [19].

Regarding terminology, "Stroking" can mean many things–stroking can be performed in many different parts of the body at various speeds and forms. As the current study aims to match effectiveness studies, unless explicitly stated otherwise, we use "stroking" to mean using one's hand to stroke another person's forearm at a speed of 1–10 cm/s, the optimal speed found in most studies. As for "C-touch", recent studies have found some C-tactile afferents in the palm skin as well, albeit at a much lower density [20]; other studies show that these afferents can be activated by deep pressure and not only by stroking, which might mean that they are activated by handholding as well [21]. As such, we avoid using the term C-touch for stroking in the rest of the manuscript as both types of touch might be activating C-tactile afferents.

The studies cited above have demonstrated many cases in which both handholding and stroking are associated with distress regulation (creating a sense of pleasantness, and reducing pain, anxiety and feelings of social exclusion). However, as detailed above, homeostatic regulation is driven by two-way feedback. While any form of touch allows for two-way sensing, the tactile sensitivity of the palms and fingers is larger by orders of magnitude than that of the forearms [22,23]. In handholding, as opposed to stroking, both participants' palms and fingers are involved, allowing both of them to sense each other with optimal sensitivity. This two-way tactile communication should help close the feedback loop required for homeostatic regulation by providing bi-directional feedback, as supported by findings showing that handholding

promotes synchrony in skin conductance response (SCR) and brain activity [7,24]. Thus, handholding may be more suitable for homeostatic regulation (i.e., regulation of intense, short-term experiences) than gentle stroking.

The importance of handholding as a bidirectional or reciprocal form of touch can also be viewed in the broader context of reciprocity in nonverbal interaction [25]. In general, people are motivated to reciprocate others' actions if those actions create pleasant arousal [26]. For example, one study found that couples are more likely to reciprocate each others' touch the more established their relationship is [27]. The use of handholding in emotion regulation is somewhat distinct from classic reciprocity as defined in the literature as the touching person is not reciprocating the touched person's *action* (i.e., they are not initiating handholding in response to touch by the other person); instead, they are responding to the other person's *emotion* in a manner which, in the case of handholding, has a reciprocal nature. As reciprocity in considered rewarding [28] there could be a general positive reaction to reciprocity; for example, in one study [29] it was found that people who performed reciprocal forms of touch were more highly regarded than people who perform non-reciprocal forms of touch.

The existing studies in the field detailed above have demonstrated the effects of handholding and stroking separately. However, studies comparing the effectiveness of one of these types of touch with the other are scarce, with studies comparing two types of touch often comparing stroking of the forearm to stroking or touching of the hand, but not to handholding. Studies that compared general pleasantness of touch between touching forearm and palm skin found little difference [30]. In one of the only studies directly comparing the emotion regulation properties of stroking of the forearm and handholding, Reddan and colleagues [24] compared participants who received painful thermal stimulation to their legs, while their romantic partners supported them via gentle stroking of the forearm, handholding or no touch. When measuring pain subjectively and through skin conductance response, no significant differences were found between handholding and gentle stroking, although effects of handholding were greater in every case. In sum, studies on the effectiveness of touch on distress regulation has shown that handholding and gentle stroking are effective forms of emotion regulation, with the few studies comparing the two finding no major differences.

While effectiveness studies attempt to explore how people experience various types of touch, they do not examine which touch people actually prefer or choose to provide, when they encounter the need to regulate emotions in real-life. Touch choice may be based on representations of touch effectiveness, but it might also have other considerations. For example, people could be choosing types of touch they have seen others perform or which were depicted culturally (i.e., in various forms of media), or types of touch which require less physical effort. Thus, effectiveness studies are not enough to understand which types of touch people choose to use. The current study aims to address this gap in the literature by examining subjective preference—presenting people with various situations and asking them which types of touch they would prefer to receive or to provide.

As detailed above, the feedback-loop and homeostatic regulation models suggests that handholding might be preferred over stroking. While we could find no studies which directly examined preference, some studies provided indirect evidence. One study examined adolescents undergoing cancer treatment and found that adolescents see handholding as an extremely effective coping method [31]. Another study which developed a scale concerning support in face of pain found that an item asking about handholding was indicative of general partner support in painful situations [32]. Notably, neither study explicitly presented stroking as a valid coping method (We could not find the initial list of items for the scale development paper by Krahé and colleagues [32]–it could be the case that one of the initial items which were discarded due to low factor loadings involved stroking). Still, this indirect evidence,

alongside the theoretical model, led us to hypothesize that handholding would be preferred over gentle stroking.

Beyond exploring our main hypothesis regarding preference, we examined several possible moderators. First, we examined whether the hypothesized preference for handholding would be stronger in more intense situations. Based on the homeostatic regulation model it was reasoned that if two-way sensory communication is indeed crucial for regulating acute disturbances in homeostasis, handholding may be more effective, and therefore preferred, as a way to regulate emotions in intense situations. Thus, we hypothesized that the preference for handholding would be stronger in intense situations.

Second, we examined in a more exploratory manner whether situation valence (i.e., whether the experience is positive or negative) and physicality (i.e., whether the experiences involved physical as opposed to purely emotional pain or pleasure) would moderate the effect. Effectiveness studies have examined the effects of touch in both positive and negative contexts (e.g., pleasantness [13] versus pain [17]), and when regulating both emotional [9] and physical [33] pain, but to the best of our knowledge have not compared these contexts to one another. As such, we did not have a specific directional hypothesis concerning valence and physicality.

Third, Study 3 aimed to examine whether including a situation that was specifically relevant to participants–namely, an injection for participants with blood/injection phobia–would induce a different pattern of results. We examined whether the level of phobia would moderate results. Finally, as we could not find effectiveness studies which performed cultural comparisons, we sought to examine whether touch preference would be different in different cultures. To do so, in Study 4 we compared the touch preferences of Arab and Jewish women. Importantly, these last two moderators–blood/injection phobia and culture–build on less established literature. As such, they were only included in one study each (Studies 3 and 4, respectively), and our investigations of them are intended to inform future research, rather than to reach solid final conclusions.

As detailed above, the aim of the current study was to connect effectiveness research with subjective preferences. While many types of stroking exist, stroking researchers have focused on gentle stroking of the forearm by the hand at a velocity of 1–10 cm/s as the optimal stroking speed; in fact, stroking at higher speeds is often used as a control condition (e.g., in a study by von Mohr and colleagues [16]). As our hypothesis was that handholding would be preferred over stroking, we sought to present stroking in the optimal way possible so that our design tests the hypothesis effectively. Therefore, in Studies 1–3 and in the second part of Study 4 we used looping videos instead of verbal labels for representing different types of touch. In the first part of Study 4, we asked participants which types of touch were provided to them during childbirth and provided standard verbal labels–"stroking" and "handholding". As both optimal and non-optimal forms of stroking would count towards "stroking" in this paradigm, we could be confident that the prevalence of optimal stroking would, if anything, be *over*estimated.

Notably, the use of videos somewhat resembles paradigms involving vicarious touch–the study of the effects of seeing people touching one another [34]. However, the current study does not ask people about their feelings while watching the videos, or their thoughts about the videos or about the people depicted in them, which are the questions explored by vicarious touch research. Instead, the current study asks participants about hypothetical or recalled scenarios and only uses the videos to describe the types of touch. While some vicarious effects might be triggered by watching the videos, the same is true for actual touch in real life: when touching or being touched by another person, we usually also see the act of touching. Additionally, to the best of our knowledge, no study of vicarious touch has compared observing handholding to observing stroking.

## Overview

In four pre-registered studies, we examined whether handholding would be subjectively preferred over gentle stroking, and whether this preference would be stronger in intense emotional situations. In each study participants rated how helpful they would find stroking or handholding in various hypothesized or recalled situations. We also examined whether participants' choice was moderated by situation intensity, touch reception versus provision, situation valence, emotional versus physical situations, and cultural differences, as detailed above.

Study 1 examined the type of touch people would prefer to *receive* from a romantic partner in hypothetical situations by comparing ratings as well as discrete choices of each type of touch. Study 2 replicated most of the results of Study 1 while correcting some methodological issues as well as examining the types of touch participants would prefer to *provide*.

Study 3 examined touch preferences in the context of a more severe stressor. Participants with some level of blood/injection phobia [35] were asked to imagine themselves receiving an injection.

Finally, in addition to hypothesized situations, Study 4 examined recalled situations and behavior. We asked women who recently gave birth to recall types of touch provided by a close person who was present during their labor, which is usually accompanied by intense pain [36], and to rate which type of touch they preferred. Study 4 also examined cultural differences between Arabic-speaking and Hebrew-speaking participants.

## Hypotheses and pre-registration

The hypotheses, study design and sample analysis code for all studies were pre-registered prior to analysis (and in the case of Study 1, prior to data collection), and final study data, methods and code were posted on the Open Science Framework (see below for links). The current study examined the following hypotheses:

1. Subjective rating (All studies): Participants will rate handholding as more helpful than stroking, especially in intense situations.

2. Subjective choice (Study 1 only): When instructed to choose one type of touch over the other, participants will prefer handholding over stroking, especially in intense situations.

3. Recollections of touch received during childbirth (Study 4 only): During childbirth, participants will have received handholding more often than stroking.

The main effect (handholding preferred over stroking overall) was pre-registered only in Studies 3 and 4. The interaction effect (handholding preferred more over stroking as intensity increases) was pre-registered in all studies. We also exploratorily examined the effects of valence, physicality, and cultural differences with no specific directional hypothesis, and exploratorily examined the effects of blood/injection phobia expecting blood/injection phobia to be associated with a stronger preference for handholding, especially in intense situations. Results regarding pre-registered hypotheses concerning loneliness in Studies 1 and 2 will be reported as part of a separate project alongside additional studies.

## General method

### Ethics statement

All procedures were done in accordance with the principles expressed in the declaration of Helsinki for the treatment of human participants. Procedures were approved by the University of Haifa Department of Psychology IRB, approval 048–21. All participants were over 18 years

old. Participants' written consent was obtained and recorded by completing a digital consent form.

## Open data and preregistration

We report how we determined our sample size, all data exclusions (if any), all manipulations, and all measures in the study. All four studies were preregistered on the Open Science Framework:

Study 1: https://osf.io/x95f4/?view_only=e272eb9ccc2a46668114685dcb8988e5

Study 2: https://osf.io/nzg6v/?view_only=b9cf1e802d894ce387c80c47ffd7bc97

Study 3: https://osf.io/8n3pu/?view_only=f23a2e3d485946ffb95d32c02e469b85

Study 4: https://osf.io/74szg/?view_only=4b12d6d2a8a04483abbd6ea0c8c225ca.

Study data, analysis code and touch videos for Studies 1–3 are available at the link below, except for touch videos for Study 4 which were not posted as participants who filmed the videos did not consent to wide distribution.

https://osf.io/nju48/?view_only=81dadc972f6545f19d107c83d8dba368

## Power analyses

Power analyses were conducted by simulation, using the R package paramtest [37]. Data were simulated with the planned number of participants and study designs, and the theoretical effect sizes detailed below. According to analyses conducted before the project began, a sample size of 100 for Studies 1 and 2 would suffice to detect a main effect with a standardized beta of .25 with very high power (Power > .999) and would have adequate power to detect an interaction effect of .0625 (Power = .812). Studies 3 and 4 were planned with a sample size of 60, with similar power (Power > .999) to detect a .25 main effect but only enough power to detect an interaction effect of .125 (Power = .86 and .904, respectively). Unfortunately, these populations proved to be difficult to recruit and sample sizes were lower than planned.

## Study 1

### Method

**Participants.** We recruited a sample of Amazon Mechanical Turk (MTurk) workers from the United States and the UK. All participants had a masters qualification and declared that they were over 18 years old, and in romantic relationships that lasted six months or more. Participants who completed the study received $1.50 in compensation. One participant did not correctly answer an attention check and was removed. Two additional participants were erroneously removed from the dataset before pre-registration as they made a technical error while submitting their questionnaires (We re-ran all analyses including these participants and all results remain the same. Results for these analyses are provided in the Robustness Tests section in the S1 Appendix).

Of 99 participants, 45 identified as female and 54 identified as male. Mean relationship length was 11.23 years (SD = 10.94). Age data was not collected in this study.

**Procedure.** All studies were conducted online via the Qualtrics survey platform. Participants began by providing informed consent and indicating the length of their romantic relationship. In the first part of the study, participants were asked to imagine themselves in 8 hypothetical situations in which their romantic partner touched them in order to make them feel better. For each situation, participants rated its intensity and positivity/negativity. They were then asked to select one type of touch which they would prefer to receive. The available types of touch were presence (i.e., no touch), handholding, and optimal stroking. To ensure

Imagine that your partner is with you in that situation, and might respond by touching you. Which kind of touch would you prefer to receive from your partner in that situation?

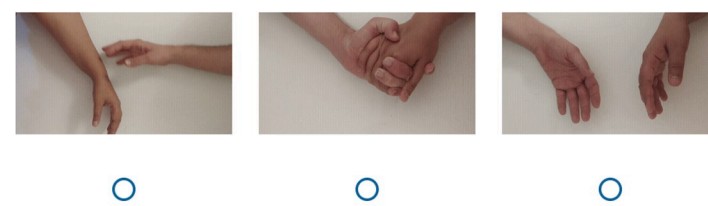

The type of touch I would prefer:

**Fig 1. The touch preference screen shown to participants.** The types of touch depicted are, from left to right, stroking, handholding, and no touch. The types of touch were depicted with videos, so the figure shows a still image from each video. Touch type order was randomized.

that the participants understood which type of touch is discussed, the types of touch were not labelled with words (e.g., "handholding"), but instead with looping 5-second videos showing the type of touch (see Fig 1, and details below).

In the second part of the study, participants saw 8 different situations, and similarly rated intensity and positivity/negativity. However, this time they were asked to provide a rating for each type of touch. Again, the types of touch were labelled with looping videos. As detailed below, each participant saw the same 16 situations which were assigned randomly to one of the two parts of the study. The touch labels were presented in random order.

After these two parts, they completed a loneliness questionnaire for a separate project.

**Measures.** *Situations.* A list of hypothetical situations was generated using a pilot study. We showed 21 participants a list of 33 emotional or physical situations generated by the researchers and asked them to rate the valence and intensity of each situation. For each context (emotional or physical), we selected the two situations rated the most positive and intense as the positive intense situations for that context. We similarly selected three additional pairs of situations (negative intense, positive not intense, negative not intense). This process yielded 16 situations classified as intense/not intense (intensity), positive/negative (valence), and physical/emotional (context). Two situations were available for each of the eight possible configurations of these variables. In the actual study, two positive situations originally designated as intense (one physical and one emotional) were each rated less intense than one of the corresponding non-intense positive situations. Thus, the designations of these two situations were switched.

In each part of the study, participants saw 8 situations. Situation order was randomized such that in each part of the study participants saw one situation of each designation (e.g., one physical, non-intense, negative situation; one physical, non-intense, positive situation, etc.).

*Situation Intensity and Valence Ratings.* Each participant rated the intensity of each situation by responding to the prompt "How intense does this situation feel?" on a 4-point Likert scale ranging from "not intense at all" to "very intense". They also rated the valence of each situation by responding to the prompt "How positive or negative does this situation feel?" on a 7-point Likert scale ranging from "very negative" to "very positive".

*Touch Preference.* In the first part of the study, touch preference was measured using a direct choice. Participants were asked "Which kind of touch would you prefer to receive from your partner in that situation?" and selected one of the types of touch on the screen. Types of touch were presented using looping videos (Fig 1).

In the second part of the study, touch preference was measured more granularly by comparing participants' ratings of each type of touch. Participants read the following instructions:

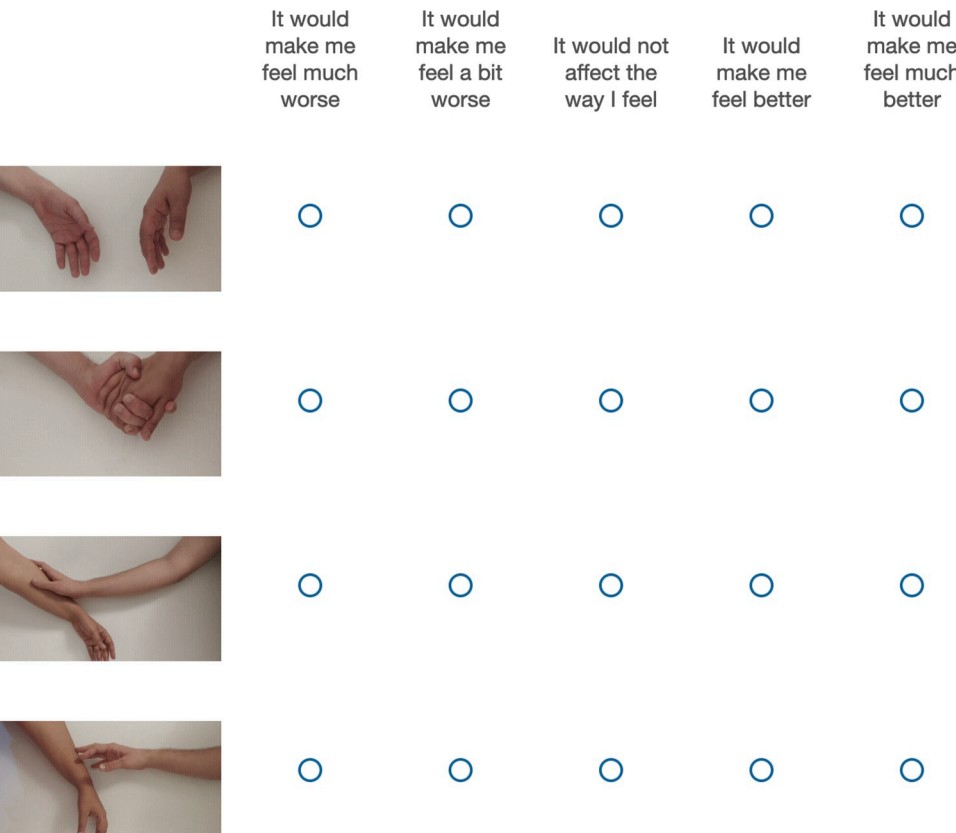

**Fig 2. The touch rating screen shown to participants.** The types of touch depicted are, from top to bottom, no touch, handholding, full palm stroking, and standard stroking. The types of touch were depicted with videos, so the figure shows a still image from each video. Note that full palm stroking was only included in Studies 2 and 3.

"Please rate how would that kind of touch affect your feeling?" and responded on a 5-point Likert scale ranging from "would make me feel much worse" to "would make me feel much better". Types of touch were presented using looping videos (Fig 2– note that this figure includes an additional type of touch which was only added in Study 2).

*Touch Label Videos.* The touch literature specifies exact parameters for optimal stroking– stroking of the arm using the index and middle fingers, at speeds of 1–10 cm/s [11,38,39]. As describing this optimal procedure could confuse participants, videos were used instead of verbal labels to depict the different types of touch (Fig 1). The videos showed only the actors' hands against a flat white background. The videos depicted presence (i.e., two hands not touching each other), stroking, and handholding. In the video depicting handholding the actors were instructed to hold their hands statically. In the stroking videos, the actors were asked to stroke their partner's arm covering an area of nine cm over a period of three seconds. The videos were then evaluated by judges to confirm that they look natural. Follow-up inspection of videos revealed a range of speeds between 4–12 cm/s.

Videos were recorded by two heterosexual couples who had been in their romantic relationship for over a year. Each couple recorded one video for each type of touch. Videos by one couple were used in the first part of the study, and videos by the other couple were used in the second part, chosen randomly for each participant.

**Statistical analysis.** All analyses were conducted using R [40]. Analyses were conducted twice, with intensity measured as a rating–the intensity of the situation as self-reported by

participants–or as a dichotomous variable–the pre-assigned intensity of the situations. For brevity, we report here only the findings for intensity rating; full results for dichotomous intensity are reported in the S2 Table in S1 Appendix. We state whenever there was a difference in significance between results using dichotomous intensity or intensity rating.

The direct touch type choices performed In the first part of the study were analyzed in a mixed logistical regression using the R package lme4 [41], using the following equation:

$$Choice_{ij} \sim b_0 + b_{1i} + b_2 * Intensity_{ij}$$

The choice of participant i in situation j–stroking or handholding was estimated using a fixed intercept ($b_0$), a random intercept for each participant ($b_{1i}$), and an intensity effect ($b_2$). Occasions on which participants chose presence over the other types of touch were not analyzed.

The touch ratings collected in the second part of the study were analyzed in a mixed linear model using the R package nlme [42], using the following equation:

$$Rating_{ij} \sim b_0 + b_{1i} + b_2 * Intensity_{ij} + b_3 * Touchtype_{ij} + b_4 * Intensity_{ij} * Touchtype_{ij}$$

The rating of participant i in situation j was estimated using a fixed intercept ($b_0$), a random intercept for each participant ($b_{1i}$), an intensity effect ($b_2$), a touch type effect ($b_3$), and an interaction effect between intensity and touch type ($b_4$). Touch type was coded 0.5 for handholding and -0.5 for stroking, so that coefficients for other effects would reflect average effects across the two touch types. In exploratory analyses with additional independent variables, fixed slopes were included for each main effect and for every possible interaction. All dependent variables were person mean-centered.

All analyses are accompanied by partial $f^2$ effect sizes, calculated using the procedures outlined by Selya and colleagues [43]. To calculate $f^2$ for a specific predictor, $R^2$ was calculated for the complete model ($R^2_{Total}$), and for the model without the predictor ($R^2_{Omitted}$); $f^2$ was calculated as ($R^2_{Total}$—$R^2_{Omitted}$) / (1-- $R^2_{Total}$).

To ensure that the effects are not due to specific statistical choices, we have re-run analyses for the main hypotheses using different statistical methods. These include modeling random slopes for all variables, using cumulative link models [44] implemented in the R package ordinal [45] (also with random slopes for all variables), and using simple repeated ANOVA tests to test effects with dichotomous intensity ratings. All of the results stayed the same–no significant effect became non-significant and vice versa. Results for the cumulative link models are provided in Tables 1 and 2, results for the other analyses are provided in the Robustness Tests section in the S1 Appendix. Note that the fully saturated cumulative link model did not converge for the second part of Study 2 (touch provision) when using participant intensity ratings (the model using dichotomous situation classifications converged successfully). Thus we ran that model without a random slope for touch type (but with random slopes for intensity and the interaction between intensity and touch type); this was the only way to have a converging model while removing only one random slope. We performed additional robustness tests on this specific study which are detailed in the S1 Appendix.

## Results

**Descriptives.**   Descriptive values for situation intensity, situation valence, and touch type rating are presented in Table 3 and in Fig 3; A figure depicting all data points is included in the S2 Fig in S1 Appendix. As for choice questions, out of 395 situations classified as intense, handholding was chosen 216 times (54.7%), stroking 87 times (22%), and no touch 92 times (23.3%). Out of 389 situations classified as non-intense, handholding was chosen 133 times

**Table 1. The effect of intensity measured by intensity ratings and touch type on touch preference analyzed using cumulative link models.**

|  |  | b(SE) | z | p |
|---|---|---|---|---|
| Study 1 | Intensity | .404(.086) | 4.69 | < .001*** |
|  | Touch Type | .698(.141) | 4.97 | < .001*** |
|  | Intensity*Touch Type | .402(.133) | 3.015 | .003** |
| Study 2 touch provision | Intensity | 0.439(.096) | 4.556 | < .001*** |
|  | Touch Type | 1.267(.205) | 6.169 | < .001*** |
|  | Intensity*Touch Type | 0.51(.128) | 3.981 | < .001*** |
| Study 2 touch reception | Intensity | 0.286(.088) | 3.262 | .001** |
|  | Touch Type | 1.484(.184) | 8.053 | < .001*** |
|  | Intensity*Touch Type | 0.495(.12) | 4.125 | < .001*** |
| Study 3 | Intensity | 1.058(.259) | 4.092 | < .001*** |
|  | Touch Type | 3.885(.627) | 6.191 | < .001*** |
|  | Intensity*Touch Type | 1.406(.339) | 4.146 | < .001*** |
| Study 1 with omitted participants | Intensity | .393(.084) | 4.705 | < .001*** |
|  | Touch Type | .651(.14) | 4.64 | < .001*** |
|  | Intensity*Touch Type | .426(.131) | 3.258 | .001** |

(34.2%), stroking 164 times (42.2%) and no touch 92 times (23.7%). These frequencies are presented in Fig 4. The slight difference in the number of situations (389 vs 395) is due to us switching the classification of two of the situations post randomization, as detailed above. The significance of none of the results regarding direct choice changed due to the switch.

**Direct choice.** We performed a Chi-square test for goodness of fit to compare the overall number of times each type of touch was chosen (349 times handholding, 251 times stroking) to a chance (equal) distribution. The test confirmed that the distribution was significantly different from chance, meaning that handholding was significantly preferred over stroking overall ($\chi2(1) = 16.007$, $p < .001$). We then performed a chi-square test for independence between the type of touch chosen and situation intensity. Touch type chosen and intensity were significantly associated with one another beyond chance (i.e., not statistically independent; $\chi2(2) = 43.317$, $p < .001$), meaning that handholding was preferred in intense situations. We tested

**Table 2. The effect of intensity measured by dichotomous situation classifications and touch type on touch preference analyzed using cumulative link models.**

|  |  | b(SE) | z | p |
|---|---|---|---|---|
| Study 1 | Intensity | .109(.115) | 0.949 | .343 |
|  | Touch Type | .694(.136) | 5.098 | < .001*** |
|  | Intensity*Touch Type | .817(.208) | 3.933 | < .001*** |
| Study 2 touch provision | Intensity | 0.072(.112) | 0.638 | .523 |
|  | Touch Type | 1.221(.193) | 6.34 | < .001*** |
|  | Intensity*Touch Type | 0.624(.186) | 3.364 | .001*** |
| Study 2 touch reception | Intensity | 0.112(.122) | 0.92 | .358 |
|  | Touch Type | 1.424(.179) | 7.939 | < .001*** |
|  | Intensity*Touch Type | 0.548(.187) | 2.927 | .003** |
| Study 3 | Intensity | 2.207(.547) | 4.033 | < .001*** |
|  | Touch Type | 3.847(.685) | 5.617 | < .001*** |
|  | Intensity*Touch Type | 2.977(.779) | 3.822 | < .001*** |
| Study 1 with omitted participants | Intensity | .093(.113) | 0.824 | .410 |
|  | Touch Type | .649(.137) | 4.74 | < .001*** |
|  | Intensity*Touch Type | .822(.204) | 4.025 | < .001*** |

**Table 3. Descriptive statistics.**

| | | All Situations | | Intense Situations | | Non-Intense Situations | |
|---|---|---|---|---|---|---|---|
| | | Person-level Mean (SD) | Within-Person SD | Person-level Mean (SD) | Within-Person SD | Person-level Mean (SD) | Within-Person SD |
| Study 1 | Intensity | 2.4(0.5) | 0.77 | 2.86(0.56) | 0.53 | 1.99(0.58) | 0.6 |
| | Valence | 4.11(0.34) | 2.12 | 3.93(0.81) | 2.24 | 4.03(0.7) | 1.79 |
| | Touch rating–– Handhold | 3.87(0.59) | 0.64 | 3.97(0.66) | 0.59 | 3.8(0.64) | 0.57 |
| | Touch rating–– Presence | 2.79(0.44) | 0.43 | 2.72(0.5) | 0.42 | 2.81(0.52) | 0.36 |
| | Touch rating–– Stroke | 3.65(0.63) | 0.63 | 3.6(0.66) | 0.65 | 3.71(0.7) | 0.54 |
| Study 2 touch reception | Intensity | 2.51(0.49) | 0.81 | 2.93(0.54) | 0.58 | 2.09(0.61) | 0.62 |
| | Valence | 4.18(0.53) | 2.05 | 4.12(0.6) | 2.22 | 4.25(0.62) | 1.82 |
| | Touch rating–Full palm stroke | 3.67(0.83) | 0.64 | 3.67(0.86) | 0.61 | 3.68(0.89) | 0.55 |
| | Touch rating–– Handhold | 4.13(0.58) | 0.6 | 4.18(0.61) | 0.58 | 4.09(0.63) | 0.51 |
| | Touch rating–– Presence | 2.8(0.57) | 0.48 | 2.75(0.61) | 0.5 | 2.86(0.6) | 0.39 |
| | Touch rating–– Stroke | 3.6(0.79) | 0.65 | 3.57(0.82) | 0.62 | 3.64(0.86) | 0.58 |
| Study 2 touch provision | Intensity | 2.38(0.54) | 0.78 | 2.77(0.61) | 0.57 | 2(0.63) | 0.6 |
| | Valence | 4.17(0.49) | 1.94 | 4.11(0.53) | 2.14 | 4.23(0.56) | 1.68 |
| | Touch rating–Full palm stroke | 3.81(0.89) | 0.65 | 3.76(0.95) | 0.64 | 3.86(0.94) | 0.57 |
| | Touch rating–– Handhold | 4.25(0.8) | 0.74 | 4.32(0.82) | 0.74 | 4.18(0.88) | 0.61 |
| | Touch rating–– Presence | 2.79(0.61) | 0.5 | 2.76(0.68) | 0.5 | 2.82(0.6) | 0.41 |
| | Touch rating–– Stroke | 3.73(0.86) | 0.64 | 3.67(0.9) | 0.69 | 3.78(0.9) | 0.52 |
| Study 3 | Intensity | 2.58(0.48) | .85 | 3.07(0.59) | .3 | 1.59(0.73) | [2] |
| | Valence | 2.82(0.79) | .98 | 2.41(1) | .52 | 3.63(1.2) | [2] |
| | MBIPI[1] | 66.94(31.06) | | | | | |
| | Touch rating–Full palm stroke | 3.14(1.05) | .65 | 3.25(1.24) | .36 | 2.9(1.08) | [2] |
| | Touch rating–– Handhold | 4.14(0.82) | .59 | 4.38(0.9) | .18 | 3.65(1.09) | [2] |
| | Touch rating–– Presence | 2.84(0.83) | .38 | 2.83(0.95) | .18 | 2.86(0.87) | [2] |
| | Touch rating–– Stroke | 3(1.1) | .6 | 3.06(1.29) | .36 | 2.88(1.07) | [2] |
| Study 4 | Intensity | 0.01(0.63) | 0.69 | 0.4(0.68) | 0.37 | -0.46(0.82) | 0.26 |
| | Touch rating–– Handhold | 62.84(28.83) | 17.09 | 64.14(28.41) | 17.88 | 64.43(31.2) | 12.91 |
| | Touch rating–– Presence | 22.86(26.22) | 8.4 | 20.46(26.1) | 5.88 | 26.1(31.09) | 7.07 |
| | Touch rating–– Stroke | 38.2(35.16) | 14.2 | 37.12(34.85) | 15.58 | 44.14(36.97) | 11.16 |

[1]Person-level variables do not differ between situations and as such have no within-person SD or situation-specific values.

[2]Study 3 had only one non-intense situation.

simple effects by comparing the distribution of touch type choices to chance, separately for intense and non-intense situations. Tests for simple effects found that in intense situations handholding was chosen more than stroking beyond chance ($\chi 2(1) = 54.921$, $p < .001$), whereas the difference between stroking and handholding in non-intense situations was not significantly different from chance ($\chi 2(1) = 3.236$, $p = .072$).

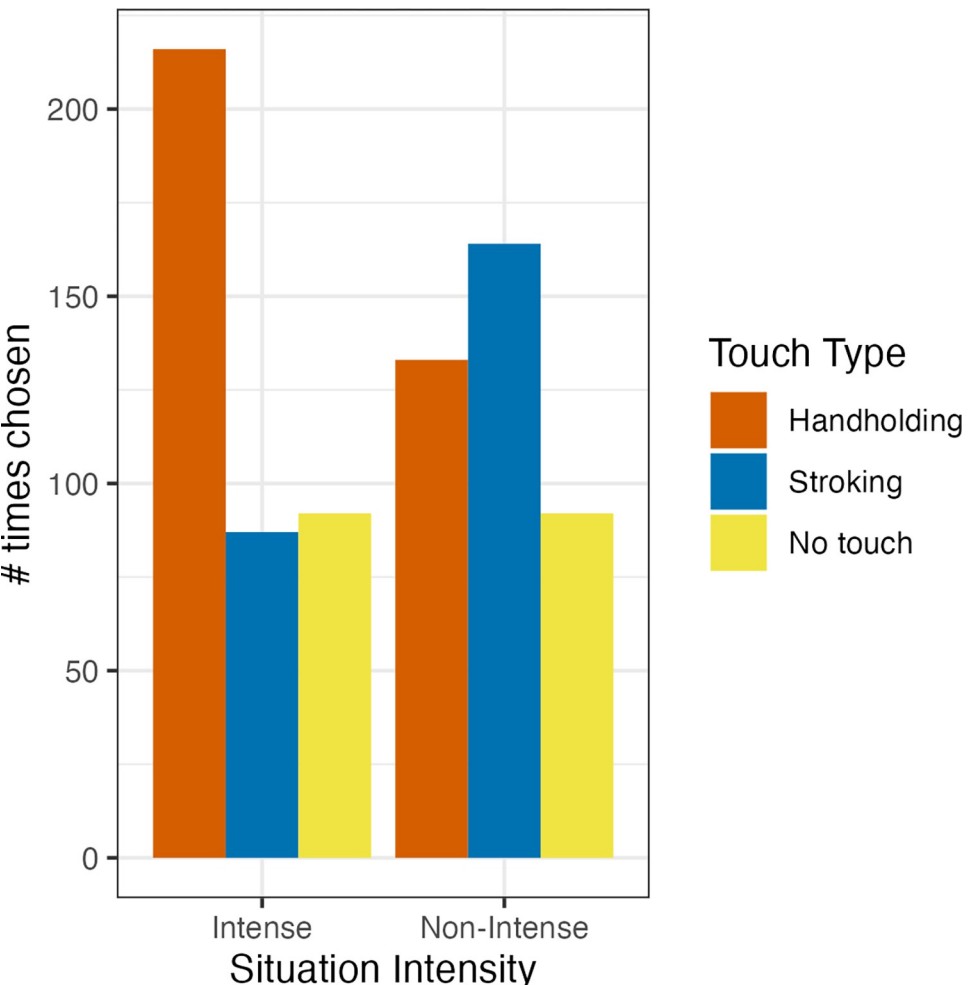

**Fig 3. Study 1 intensity and touch type rating frequencies by situation intensity.** The figure shows how many times each type of touch was chosen as the preferred type of touch for a situation.

To examine whether participants would usually choose handholding over stroking, especially in situations which they rated as more intense (as opposed to the Chi-square tests which compared preferences between situations which were pre-classified as intense or not intense), we conducted a logistic mixed linear regression using the R package lme4 [41] to examine the effect of intensity on touch choice, as detailed above (situations in which "no touch" was chosen were removed). The results confirmed our hypothesis that handholding would be chosen significantly more often stroking (i.e., the intercept was positive and significant; b = .329(SE = .096), z = 3.427, p < .001) and that this effect would be stronger in more intense situations (i.e., an intensity effect, b = .593(SE = .101), z = 5.847, p < .001). As a robustness check, we also ran the model with saturated random slopes. Again, the results confirmed our hypothesis that handholding would be chosen significantly more often stroking (i.e., the intercept was positive and significant; b = .302(SE = .105), z = 2.861, p = .004) and that this effect would be stronger in more intense situations (i.e., an intensity effect, b = .77(SE = .16), z = 4.809, p < .001). This pattern of results also held when including the two omitted participants.

**Rating.** To examine whether participants would rate handholding as better at regulating their emotions than stroking, especially in situations which they rated as more intense, we

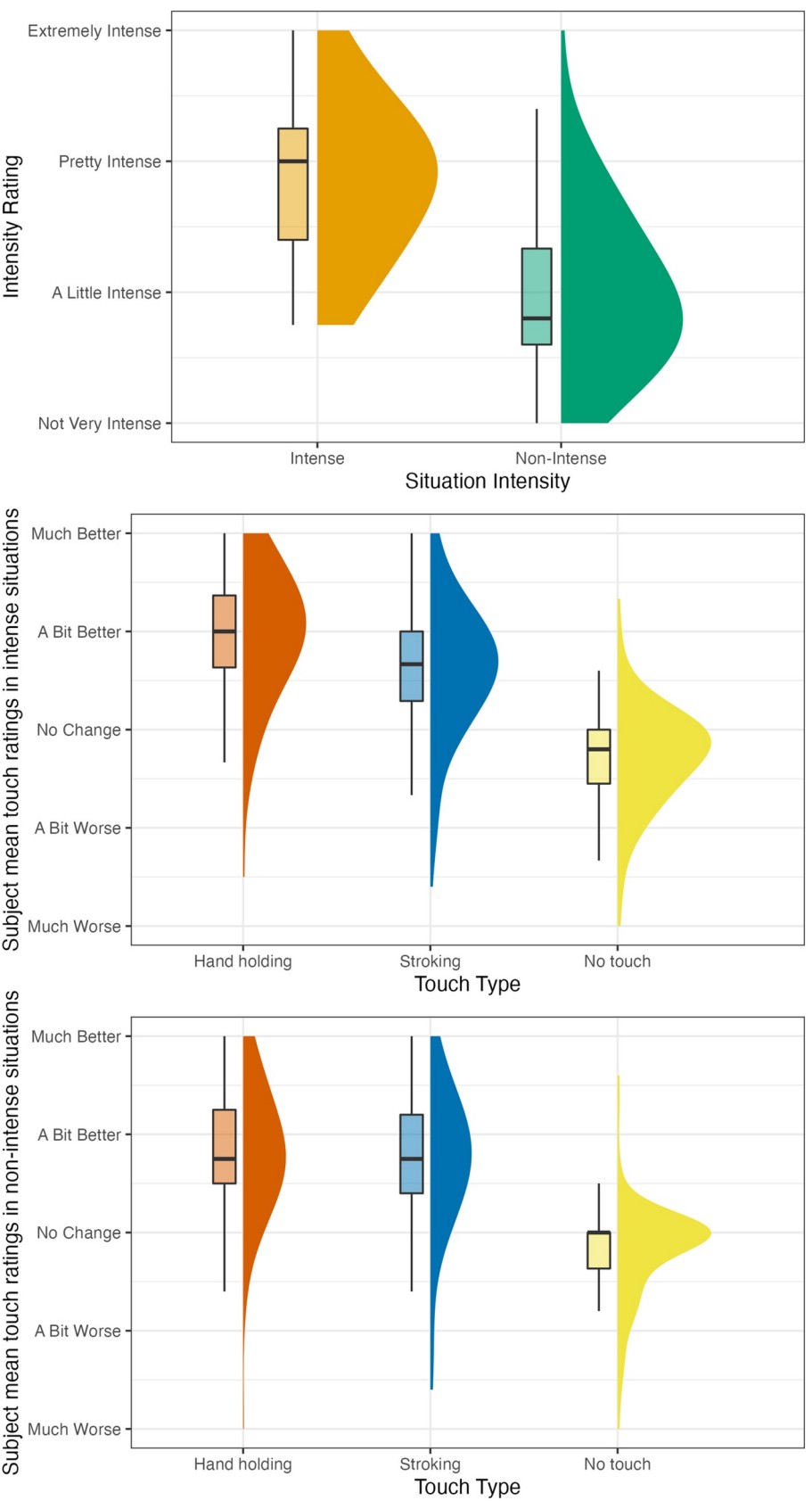

**Fig 4. Number of participants who chose each type of touch by situation intensity.** The figure plots the distribution of subject mean ratings for each type of touch for each situation (e.g., the mean of one specific subject's ratings of stroking for 4 intense situations would be one data point). Touch ratings refer to the way participants thought their feelings would change if they received this type of touch.

conducted a mixed linear regression analysis as detailed above. Handholding was rated significantly higher than stroking, and the interaction between intensity and touch type was significant such that the difference between handholding and stroking was even larger in more intense situations (Full numerical results are provided in Table 4; results are demonstrated in Fig 5).

We performed simple slope analyses via contrasts using the simple_slopes function of the R package reghelper [46]. Handholding was rated significantly higher than stroking at both high and low intensity levels, i.e., one standard deviation above and below mean intensity (when using dichotomous intensity classification, for intensity one standard deviation below the mean, the difference was in the same direction but not significant, p = .059). Ratings for handholding significantly increased with situation intensity. Ratings for stroking did not significantly change with intensity (when using dichotomous intensity classification, ratings for stroking significantly *decreased* with intensity). Full simple slope results are reported in Table 5.

**Exploratory analyses.** We performed exploratory analyses on moderation of these effects by valence and physicality in the second part of the study. Participants rated all types of touch higher in negative situations than in positive situations, especially when those situations were physical. The difference between participants' ratings of handholding as opposed to stroking was larger in positive than in negative situations and larger in emotional as opposed to physical situations. Full analysis tables are provided in the S1 Appendix.

**Table 4. The effect of situation intensity as measured by participant ratings and of touch type on touch rating.**

|  |  | b(SE) | 95% CI | t(df) | p | f² |
|---|---|---|---|---|---|---|
| Study 1 | Intercept | 3.759(.057) | 3.65,3.87 | 66.231(1482) | < .001*** | 0 |
|  | Intensity | 0.117(.022) | 0.07,0.16 | 5.2(1482) | < .001*** | .011 |
|  | Touch Type | 0.226(.035) | 0.16,0.30 | 6.375(1482) | < .001*** | .016 |
|  | Intensity*Touch Type | 0.127(.045) | 0.04,0.22 | 2.829(1482) | .005** | .003 |
| Study 2 touch provision | Intercept | 3.886(.066) | 3.76,4.02 | 58.613(2320) | < .001*** | 0 |
|  | Intensity | 0.105(.02) | 0.07,0.14 | 5.359(2320) | < .001*** | .007 |
|  | Touch Type | 0.389(.031) | 0.33,0.45 | 12.401(2320) | < .001*** | .036 |
|  | Intensity*Touch Type | 0.12(.039) | 0.04,0.20 | 3.047(2320) | .002** | .002 |
| Study 2 touch reception | Intercept | 3.885(.067) | 3.75,4.02 | 58.129(2320) | < .001*** | 0 |
|  | Intensity | 0.062(.02) | 0.02,0.10 | 3.104(2320) | .002** | .002 |
|  | Touch Type | 0.495(.033) | 0.43,0.56 | 15.101(2320) | < .001*** | .055 |
|  | Intensity*Touch Type | 0.113(.04) | 0.04,0.19 | 2.859(2320) | .004** | .002 |
| Study 3 | Intercept | 3.603(.115) | 3.38,3.83 | 31.34(405) | < .001*** | 0 |
|  | Intensity | 0.193(.052) | 0.09,0.30 | 3.74(405) | < .001*** | .019 |
|  | Touch Type | 1.069(.092) | 0.89,1.25 | 11.636(405) | < .001*** | .18 |
|  | Intensity*Touch Type | 0.249(.103) | 0.05,0.45 | 2.404(405) | .017* | .008 |
| Study 4 | Intercept | 51.108(4.677) | 41.88,60.34 | 10.926(185) | < .001*** | .001 |
|  | Intensity | 3.155(2.378) | -1.54,7.85 | 1.327(185) | .186 | 0 |
|  | Touch Type | 22.627(3.456) | 15.81,29.45 | 6.546(185) | < .001*** | .1 |
|  | Intensity*Touch Type | 0.729(4.686) | -8.52,9.97 | 0.156(185) | .877 | 0 |

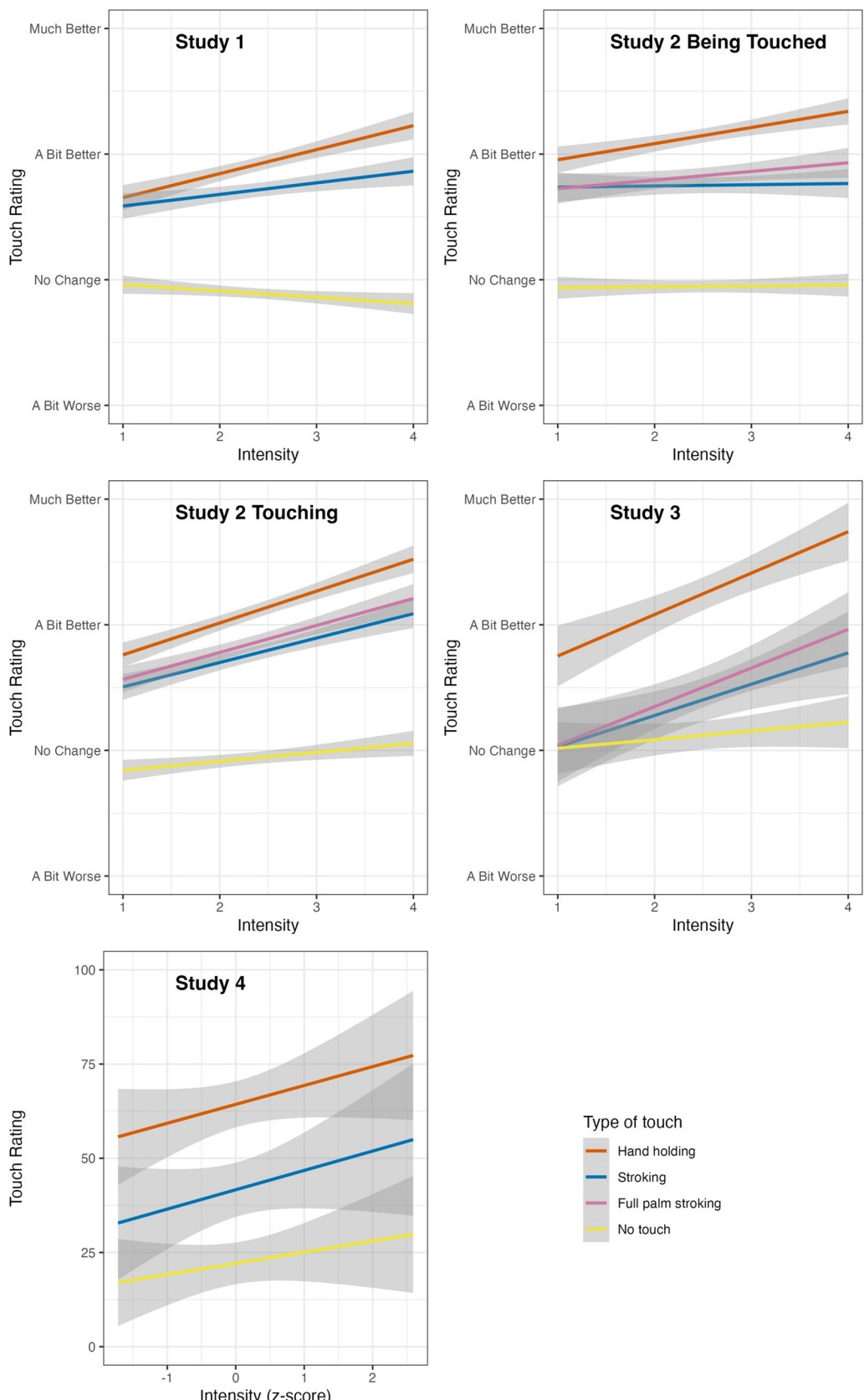

**Fig 5. Participants' touch ratings as predicted by situation intensity and type of touch.** The figure shows the regression results for the association between the ratings for each type of touch and situation intensity.

## Study 2

Study 2 aimed to replicate the results of Study 1, while also examining the types of touch participants thought would be better to *provide* to their partners, and controlling for the amount of skin touching by including a full palm stroking touch type.

## Method

**Participants.** We recruited participants similarly to Study 1, but included workers from all countries. We attempted to recruit 100 participants as per the pre-registration. One participant did not ask for payment through the MTurk platform, resulting in 101 valid entries. Of 101 participants, 49 identified as female and 52 identified as male. Mean relationship length was 11.04 years (SD = 10.41). Age data was not collected in this study. Participants received compensation as in Study 1. All participants completed all of the measures.

**Procedure.** The procedure was similar to that of Study 1. Study 2 was also divided into two parts. The first part was identical to the second part of Study 1. Participants imagined themselves in eight situations and rated to what extent receiving each type of touch would change their feelings. The second part was the same, but participant were asked to imagine the various situations happening to their partner, and rated the extent to which providing each type of touch would change their partner's emotions.

**Measures.** Intensity and valence measures were identical to Study 1. The situations used were identical to Study 1, using the revised situation designation (i.e., with two pairs of situations already switched). An additional type of touch was added to both parts of the study–– full palm stroking, which is identical to the stroking video except that the actors used their entire hand instead of two fingers, thus matching the amount of skin touching in the handholding video.

**Table 5. Simple slope analyses for the interaction between intensity measured as a self-reported rating and touch type in Studies 1–3.**

| | | b(SE) | t(df) | p |
|---|---|---|---|---|
| Study 1 | Higher ratings for handholding (vs. stroking) at low intensity (-1 SD) | 0.13(0.05) | 2.51(1482) | .012* |
| | Higher ratings for handholding (vs. stroking) at high intensity (+1 SD) | 0.33(0.05) | 6.51(1482) | < .001*** |
| | Higher ratings for handholding as intensity increases | 0.18(0.03) | 5.68(1482) | < .001*** |
| | Higher ratings for stroking as intensity increases | 0.05(0.03) | 1.68(1482) | .094† |
| Study 2 touch provision | Higher ratings for handholding (vs. stroking) at low intensity (-1 SD) | 0.29(0.04) | 6.61(2320) | < .001*** |
| | Higher ratings for handholding (vs. stroking) at high intensity (+1 SD) | 0.48(0.04) | 10.92(2320) | < .001*** |
| | Higher ratings for handholding as intensity increases | 0.17(0.03) | 5.15(2320) | < .001*** |
| | Higher ratings for stroking as intensity increases | 0.05(0.02) | 2.00(2320) | .045* |
| Study 2 touch reception | Higher ratings for handholding (vs. stroking) at low intensity (-1 SD) | 0.40(0.05) | 8.65(2320) | < .001*** |
| | Higher ratings for handholding (vs. stroking) at high intensity (+1 SD) | 0.59(0.05) | 12.70(2320) | < .001*** |
| | Higher ratings for handholding as intensity increases | 0.12(0.03) | 3.65(2320) | < .001*** |
| | Higher ratings for stroking as intensity increases | 0.00(0.02) | 0.21(2320) | .832 |
| Study 3 | Higher ratings for handholding (vs. stroking) at low intensity (-1 SD) | 0.85(0.13) | 6.52(405) | < .001*** |
| | Higher ratings for handholding (vs. stroking) at high intensity (+1 SD) | 1.29(0.13) | 9.92(405) | < .001*** |
| | Higher ratings for handholding as intensity increases | 0.32(0.08) | 3.76(405) | < .001*** |
| | Higher ratings for stroking as intensity increases | 0.07(0.06) | 1.16(405) | .248 |

## Results

**Descriptives.**   Descriptive values for situation intensity, situation valence, and touch type rating are presented in Table 3 and Figs 6 and 7; Figures depicting all data points are included in the S3 and S4 Figs in S1 Appendix.

**Touch Reception–- Rating.**   As in Study 1, to examine whether participants would rate handholding as better at regulating their emotions than stroking, especially in situations which they rated as more intense, we conducted a mixed linear regression analysis as detailed above. Preliminary analyses found no significant difference between partial stroke and full-palm stroke videos, and thus they were both coded as stroking in analyses.

Handholding was rated significantly higher over stroking, and the interaction between intensity and touch type was significant. Again, the difference between handholding and stroking was even larger in more intense situations (Full numerical results are provided in Table 3; results are demonstrated in Fig 5). As in Study 1, simple slope analyses via contrasts found that (1) handholding was rated significantly higher than stroking at both high and low intensity levels, i.e., one standard deviation above and below mean intensity and touch ratings for handholding, but not for stroking, significantly increased with intensity (this last finding–ratings for handholding increasing with intensity–was in the same direction but not significant when intensity was classified dichotomously, p = .088). Full simple slope results are reported in Table 4.

**Touch Provision–Rating.**   As for touch reception, handholding was rated significantly higher than stroking, the interaction between intensity and touch type was significant. Again, the difference between handholding and stroking was even larger in more intense situations (Full numerical results are provided in Table 3; results are demonstrated in Fig 5). Simple slope analyses via contrasts found that (1) handholding was rated significantly higher than stroking at both high and low intensity levels, i.e., one standard deviation above and below mean intensity and (2) touch ratings for both handholding and stroking, significantly increased with intensity (when intensity was classified dichotomously ratings for stroking significantly *decreased* with intensity). Full simple slope results are reported in Table 4.

**Exploratory analyses.**   We performed exploratory analyses on moderation of these effects by valence and physicality. For touch reception, participants rated all types of touch higher in negative situations than in positive situations, especially when those situations were physical (when intensity and valence were rated dichotomously, the difference between negative and positive situations was also larger in intense as opposed to non-intense situations). The difference between participants' ratings of handholding as opposed to stroking was larger in emotional as opposed to physical situations.

For touch provision, participants rated all types of touch higher in emotional as opposed to physical situations, especially when those situations were positive (when intensity and valence were rated dichotomously, participants also rated all types of touch higher in negative as opposed to positive situations, and the effects of valence and physicality were stronger when situations were intense). The difference between participants' ratings of handholding as opposed to stroking was larger in emotional as opposed to physical situations (this was not significant when intensity and valence were rated dichotomously; instead, the difference between ratings of handholding and striking was larger in negative as opposed to positive situations). Finally, the difference between participants' ratings of handholding as opposed to stroking was larger in negative and intense situations than in positive, non-intense ones (this was not significant when intensity and valence were rated dichotomously; instead, the difference between ratings of handholding and stroking was larger in emotional and intense situations than in physical and non-intense ones). Full analysis tables are provided in the S1 Appendix.

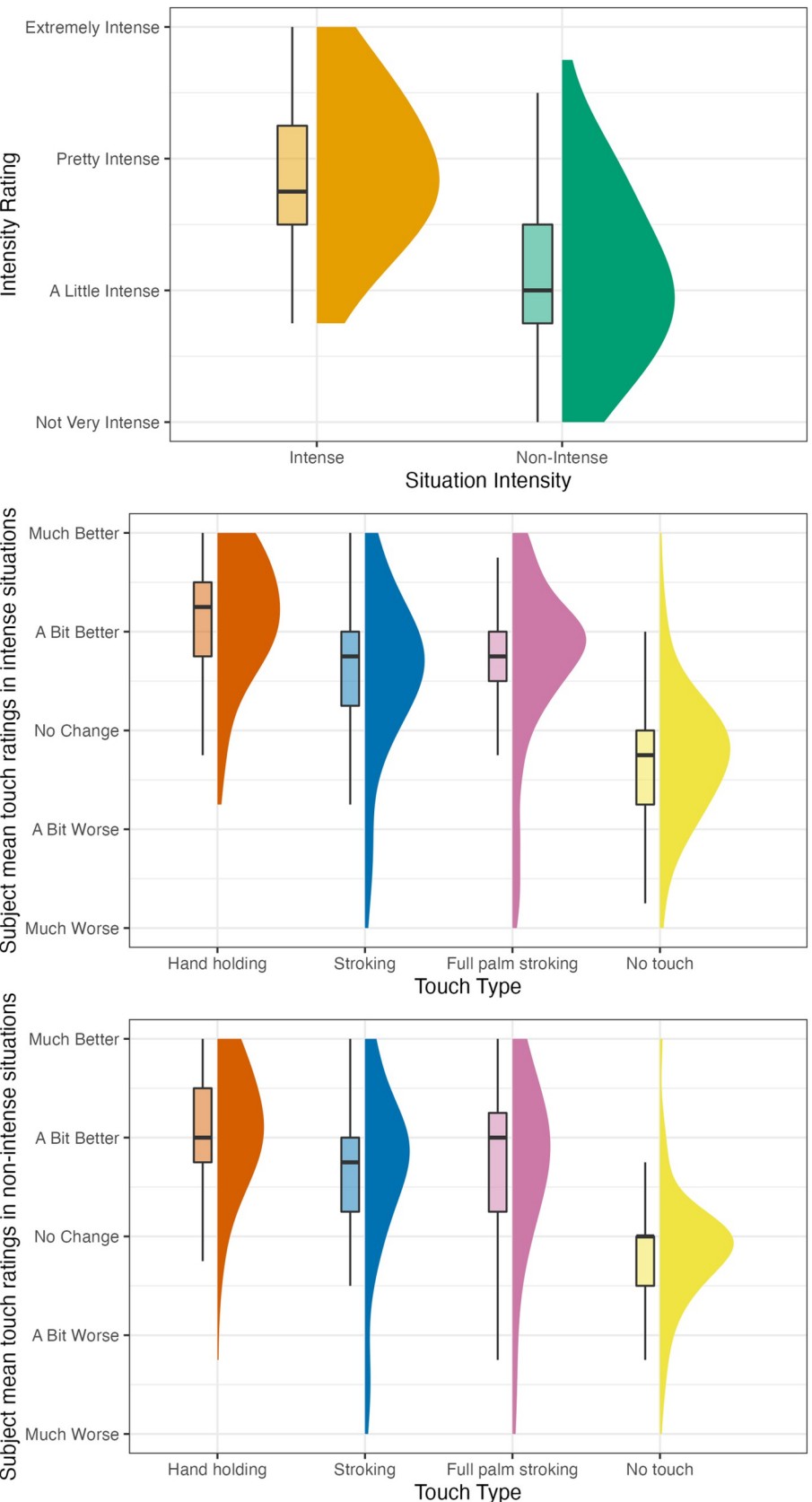

**Fig 6. Study 2 part 1 (touch reception) intensity and touch type rating frequencies by situation intensity.** The figure plots the distribution of subject mean ratings for each type of touch for each situation (e.g., the mean of one specific subject's ratings of stroking for 4 intense situations would be one data point). Touch ratings refer to the way participants thought their feelings would change if they received this type of touch.

## Study 3

Study 3 aimed to replicate the results of the previous studies using a situation which was known to be relevant to participants, by recruiting participants with blood/injection phobia and asking them about touch preferences while receiving an injection (or a control situation).

### Method

**Participants.** In Study 3 we recruited Hebrew-speaking participants over social media. All participants declared they were over 18 years old, in a romantic relationship lasting over six months, and had some fear of injections or blood. Twenty-one participants were recruited as volunteers. Because recruitment was slow, we added a payment equivalent to $6.00 and recruited an additional 34 participants. Four participants did not complete the entire questionnaire, three of which coded one out of three situations and one coded two out of three situations. Two participants of these four were erroneously not mentioned in the pre-registration document. All analyses used only the 51 participants for whom full data is available.

Of the 51 participants who completed the questionnaire, 32 identified as female and 19 identified as male. Mean relationship length was 6.85 years (SD = 6.84). Age data was not collected in this study. Participants' mean score on the Multidimensional Blood/Injection Phobia Inventory (MBIPI; see S1 Appendix) was 66.941, slightly higher than the mean previously found in a clinical sample (61.4; [47]). Thus, the group had clinically relevant levels of blood/injection phobia. All participants completed all of the measures.

**Procedure.** The study was modeled after the second part of Study 1 and the first part of Study 2. Participants were asked to rate how the four types of touch examined in Study 2 would change their feelings in three medically relevant situations (see measures for details). At the end of the study they completed a blood/injection phobia questionnaire.

**Measures.** Measures for intensity, valence and touch rating were identical to Study 2, but translated to Hebrew.

*Situations.* Instead of the situations used in Studies 1 and 2, three different situations were presented: getting a vaccination shot, getting a blood test, and getting height and weight measured. For dichotomous intensity analyses (detailed in the S1 Appendix), the first two situations were considered intense and the last was considered not intense.

*Blood\Injection Phobia.* The level of blood\injection phobia was measured in Study 3 using the Multidimensional Blood/Injury Phobia Inventory (Wenzel & Holt, 2003), which asks participants to rate the extent to which they agree with each one of 40 statements on a 5-point Likert-type scale (from "not at all" to "completely").

### Results

**Descriptives.** Descriptive values for situation intensity, situation valence, touch type rating and blood/injection phobia are presented in Table 3 and in Fig 8; A figure depicting all data points is included in the S5 Fig in S1 Appendix.

**Touch Reception–Rating.** As in Study 2, handholding was rated significantly higher than stroking, and the interaction between intensity and touch type was significant (Full numerical results are provided in Table 3; results are demonstrated in Fig 5). The difference between handholding and stroking was even larger in more intense situations. Again, simple slope

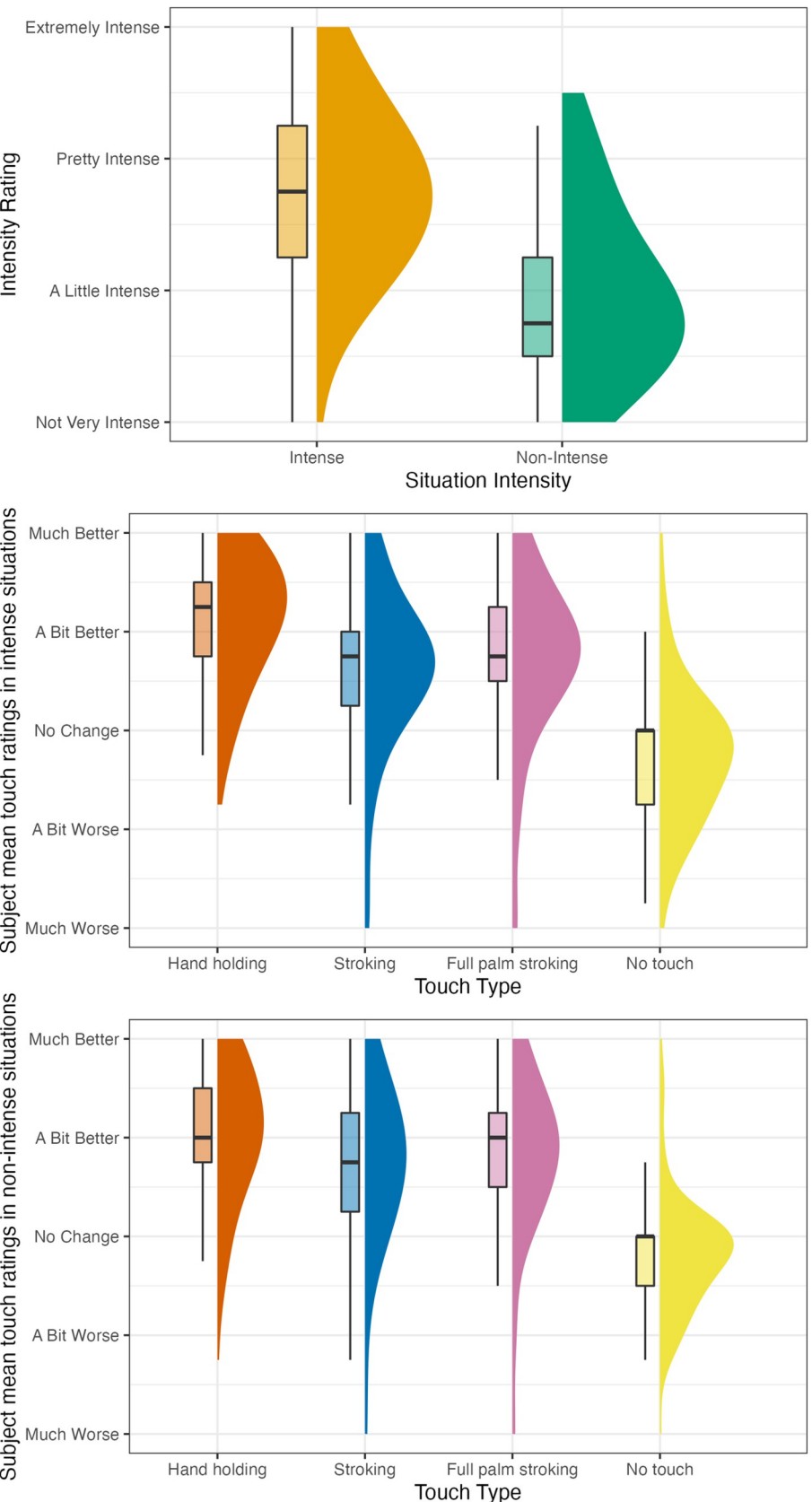

**Fig 7. Study 2 part 2 (touch provision) intensity and touch type rating frequencies by situation intensity.** The figure plots the distribution of subject mean ratings for each type of touch for each situation (e.g., the mean of one specific subject's ratings of stroking for 4 intense situations would be one data point). Touch ratings refer to the way participants thought their partners' feelings would change if they received this type of touch.

analyses via contrasts found that handholding was rated significantly higher than stroking at both high and low intensity levels, i.e., one standard deviation above and below mean intensity (Full results in 2). Ratings for handholding, but not for stroking, significantly increased with intensity (when intensity was classified dichotomously, ratings for stroking significantly *increased*).

**Exploratory analyses.** We performed exploratory analyses on moderation of these effects by valence and by blood/injection phobia. No effects were found for valence. Contrary to our expectations, higher levels of blood/injection phobia were associated with a *weaker* preference for handholding over stroking; However, simple slope analyses revealed that the preference for handholding remained significant even when injection phobia was 2 standard deviations above the mean (b = .479(SE = .203), t(402) = 2.354, p = .019). Blood/injection phobia levels did not moderate the effects of situation intensity on touch type preference. Full analysis tables are provided in the S1 Appendix.

## Study 4

Study 4 aimed to replicate the previous studies using a new situation–childbirth. Instead of hypothetical scenarios, participants answered questions about their recalled experiences while they were giving birth.

### Method

**Participants.** We used social media to recruit 20 Arabic-speaking and 25 Hebrew-speaking participants (i.e., Arabic and Hebrew as their mother tongue, respectively). In Israel, where the study was performed, Hebrew native speakers are overwhelmingly Jewish while Arabic native speakers are overwhelmingly Arab [48], making the use of language as a proxy for culture a common practice in research on these cultural groups (e.g., [49,50]). Participants declared that they were over 18 years old and had given birth during the previous two months. All participants were female. Four Arabic-speaking and five Hebrew-speaking participants completed the questionnaires quite inadequately, with none of the dependent variables, and were removed from the study. Participants' ages ranged from 22 to 41, with a mean of 30.8 years (SD = 5.03). Relationship length was not examined in this study.

**Procedure.** After providing informed consent, participants rated the amount of physical pain they experienced during childbirth, rating separately the amount of pain they experienced during contractions and between contractions. They were then asked which types of touch had been provided by a person close to them (e.g., partner or family member) who was present during their labor and delivery. They were then asked to evaluate the extent to which each type of touch helped reduce their physical pain, again answering separately about their experience during contractions, and then between contractions. When a type of touch was not actually provided, they were instructed to rate how they thought it would have affected them.

Then, they reported the amount of *emotional* pain they experienced, and rated the extent to which each type of touch helped reduce their *emotional* pain, again answering separately about their experience during contractions, and then between contractions.

All participants answered questions concerning recalled touch and concerning intensity and the effects of touch on physical pain during contractions, and concerning the intensity of

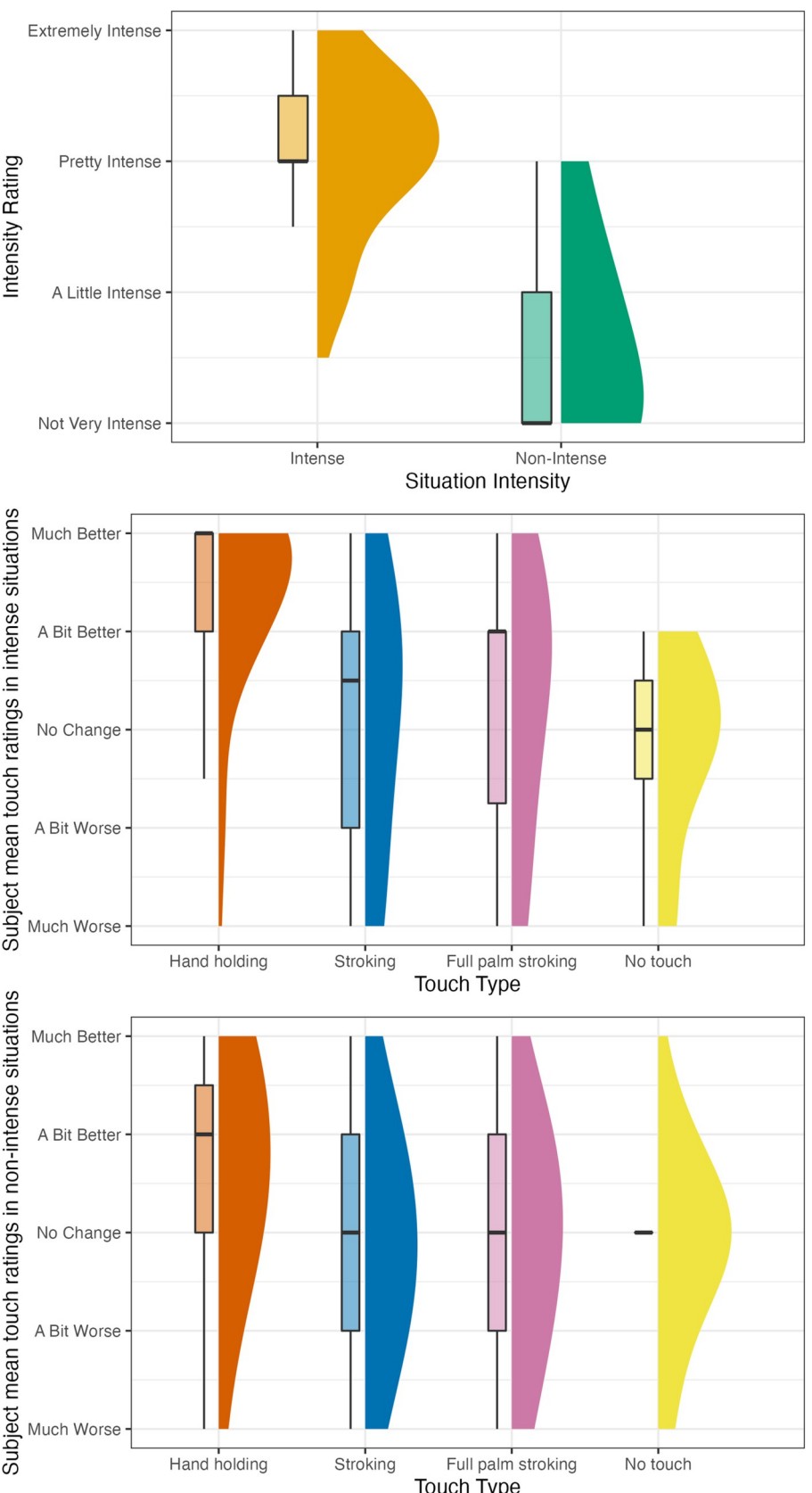

**Fig 8. Study 3 intensity and touch type rating frequencies by situation intensity.** The figure plots the distribution of subject mean ratings for each type of touch for each situation intensity (e.g., the mean of one specific subject's ratings of stroking for the 2 intense situations would be one data point). Touch ratings refer to the way participants thought their feelings would change if they received this type of touch.

physical pain between contractions. 7 participants quitted the survey at that point, leaving 29 participants who answered questions about touch effects on physical pain between contractions. One participant quitted the study at that point, leaving 28 participants who answered questions on intensity and touch effects on emotional pain during contractions. 9 participants quitted the survey at that point, leaving 19 participants who completed the final questions on emotional pain between contractions (We could not find any meaningful differences between participants who quit early and participants who did not; The main effect found for touch—higher ratings for handholding over stroking—was found even when looking only on touch effects on physical pain during contractions, where data was available for all participants).

Measures. **Situation Intensity Rating**

Intensity was measured using participants' rating of physical pain as measured on a 0–100 sliding scale and their rating of emotional pain as measured using five items selected from the State-Trait Personality Inventory (STPI; [51]). The measure demonstrated high reliability, with Cronbach's alpha of .89. Values were normalized to a mean of 0 and a standard deviation of 1 so that physical and emotional intensity would be comparable.

## Touch Preference

At the beginning of the study, participants were asked which types of touch a person close to them who was present at childbirth provided. The options were presented verbally, and included handholding, stroking, touching the lower back, hugging, no touch, or "other"; participants could select one, multiple, or no types of touch. When participants were asked to rate types of touch, they were asked to rate the extent to which each of the aforementioned types of touch reduced their pain–physical or emotional–on a 1–100 scale, with no verbal indicators for endpoints. For example, for physical pain the prompt was "For each of the following types of touch, please rate on a scale of 1 to 100 how much did it help reduce your physical pain during the contractions". Touch types were presented using videos, recorded by a different couple than the actors for the previous studies' videos. These videos showed actors' full bodies below the heads, wearing nondescript clothes on a white background. Videos were included for each of the aforementioned types of touch (handholding, stroking, touching the lower back, hugging, or no touch). There was no full palm stroking video in this study. Data on types of touch other than handholding and stroking was not analyzed for the current manuscript as they were only measured in Study 4 and no hypotheses including them were pre-registered.

## Results

**Descriptive statistics.** Descriptive values for situation intensity, situation valence, and touch type rating are presented in Table 3 and in Fig 9; A figure depicting all data points is included in the S6 Fig in S1 Appendix.

**Recalled perceptions of touch during childbirth.** Out of 36 participants, 7 (19.4%) reported that the helping person stroked them, while 25 (69.4%) reported that the person held their hand. To confirm our hypothesis that handholding would be more prevalent than stroking, we performed a chi-square for goodness of fit test comparing the actual distribution between handholding and stroking (19.4% and 69.4%) with a chance distribution (50% for each type of touch). The test revealed that the distribution was significantly different from

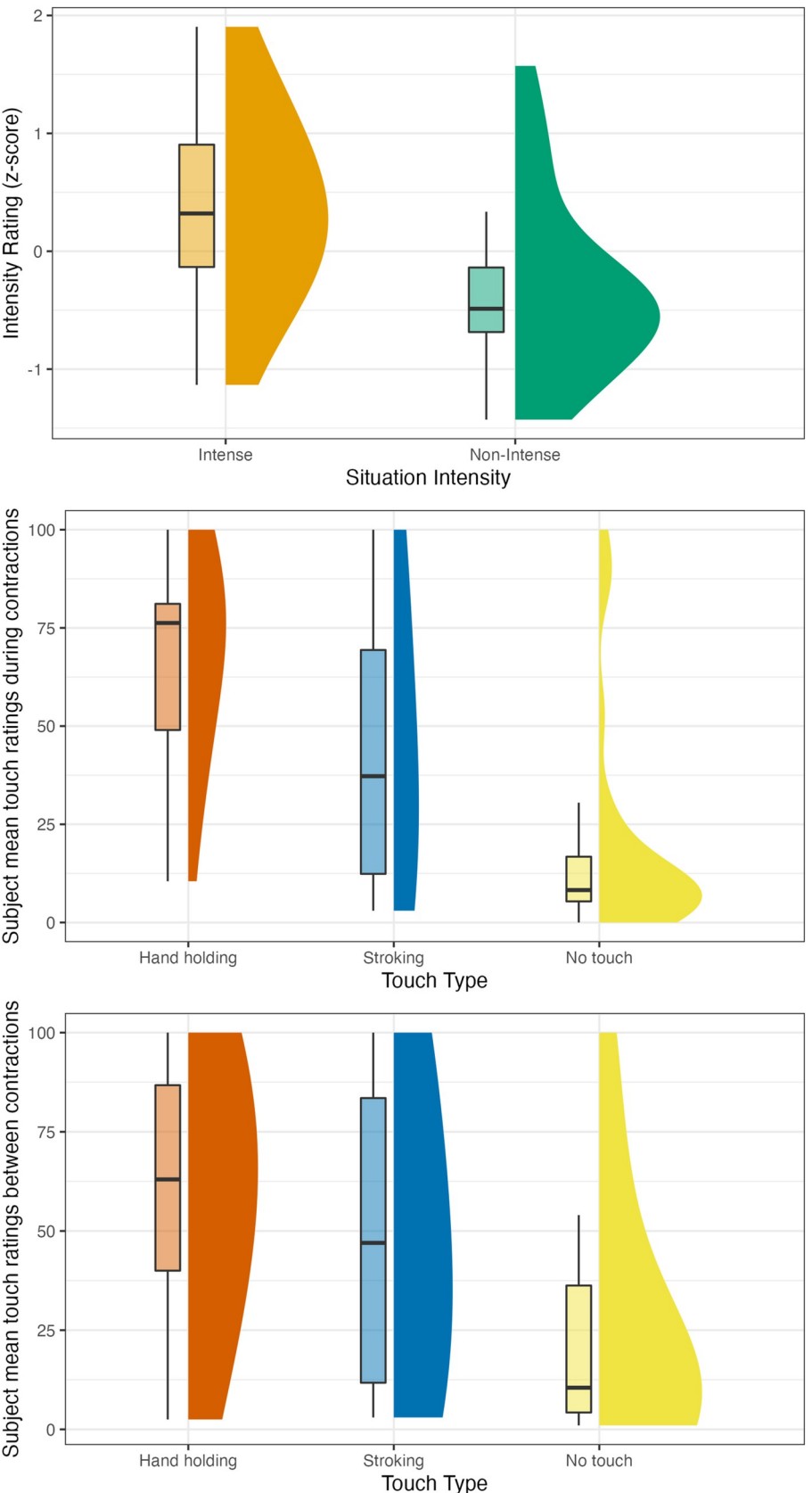

**Fig 9. Study 4 intensity and touch type rating frequencies by situation.** The figure plots the distribution of subject mean ratings for each type of touch for each situation intensity (e.g., the mean of one specific subject's ratings of stroking during contractions would be one data point). Touch ratings refer to the way participants thought their feelings would change if they received this type of touch.

chance ($\chi^2(1)$ = 10.125, $p$ = .001). Importantly, "stroking" was not explicitly specified to be on the hand or forearm. Thus, out of the 7 participants who reported receiving stroking, some may have received stroking in other parts of the body, making the difference in prevalence between handholding and stroking of the hand or forearm even larger.

**Touch Reception–Rating.**   As in previous studies, handholding was rated significantly higher over stroking. However, the interaction between intensity and touch type was in the hypothesized direction but was not significant (Full numerical results are provided in Table 3; results are demonstrated in Fig 4).

**Exploratory analyses.**   We performed exploratory analyses on moderation of these effects by valence, physicality and culture. Participants rated all types of touch higher in emotional, as opposed to physical situations. Arab-speaking participants rated all types of touch higher than Hebrew-speaking participants. No other effects were found. Full analysis tables are provided in the S1 Appendix.

## Discussion

The overall pattern of results confirmed our hypotheses. Handholding was preferred over stroking, and in almost all studies that preference was stronger in hypothetical and recalled situations judged to be intense. Handholding was preferred by both recipients and providers and was strongly preferred behaviorally by people assisting women giving birth (as recalled by the women). These results suggest that, at least in hypothetical and recalled situations, participants subjectively preferred handholding over stroking as a type of touch suitable for regulation of acute emotions and pain.

### Overall preference for handholding

In all four studies, using a variety of populations and situations and several methodologies, handholding was preferred over stroking as a form of emotion regulation. These findings support the feedback loop theory [6] which suggests that a key part of homeostatic emotion regulation via touch is establishing two-way communication which allows the toucher to optimally react to the other person's emotional state. In handholding, both people are using their palms–one of the most sensitive parts of the skin [22]–allowing them to establish high quality tactile communication. While the single study we could find comparing effectiveness of handholding and stroking did not find significant differences [24], that could have been due to some specific circumstances of that study, or due to its limited sample size.

However, it could also be the case that the preference for handholding is unrelated to differences in effectiveness. One possibility is that as handholding is more prevalent in society, it is more recognizable and enjoys a familiarity effect [52]. Holding hands is ubiquitous and even appears in some monkey populations [53], and features in a variety of cultures: for example, it is mentioned in well known Western cultural texts [54,55], is a popular public gesture of affection in Vietnamese culture [56], and features in indigenous Taiwanese dances [57].

Another possibility is that top-down processes might lead to handholding being thought of as more effective. While, as discussed in the introduction, bottom-up processes such as the activation of CT fibers [58] may be more dominant in stroking, top-down processes have been identified in both types of touch. For example, studies have shown that participants'

relationship satisfaction moderated the effects of handholding, such that participants with more satisfying relationships exhibited stronger emotion regulation effects [9,59]; another study has shown that participants' attachment style moderated the pleasantness caused by stroking, such that participants with more anxious attachment styles experienced less pleasantness. While we could find no direct comparison of top-down effects, one study [29] explored difference in the ways people perceive reciprocal types of touch, which include handholding, as opposed to non-reciprocal types of touch, which include stroking (although these two specific types of touch were not directly examined in the study). The study showed that people regarded both participants in reciprocal touch slightly higher (on dimensions of valence and likability) than *providers* of non-reciprocal touch, and significantly higher than *recipients* of non-reciprocal touch. Thus, people thinking about providing or receiving handholding might be thinking about themselves as more likeable and with a more positive valence, which might lead them to choose this type of touch over others regardless of its actual bottom-up effects.

Finally, other possible mechanisms include differences between active and passive touch and the regulation effects of holding an object (not necessarily a hand). Regarding active and passive touch, to the best of our knowledge no existing study examined differences in their emotion regulation capacities. However a review of differences in their *sensory* capabilities found studies pointing to either type of touch as better performing [60]. The authors of the review suggest that the differing factor may be task complexity, with passive touch performing better in sensing simple (as opposed to complex) stimuli. It could be the case that another person's skin is not an overly complex stimulus, and as such the passive touch involved in handholding might perform better. Additionally, in both handholding and stroking the touch recipient is engaged in passive touch, limiting differences in this regard to the touch provider.

Regarding the effects of holding an object, it is theoretically possible that the effects of handholding are merely the effects of holding an object. However, while holding an object (e.g., a squeeze ball) is a recognized form of distraction in the medical literature, it is not significantly different than other forms of distraction such as watching cartoons or hearing music [61,62]. Thus, there is no specific reason to believe that stroking would not generate similar distraction. Indeed, emotion regulation studies directly comparing holding an object to handholding find larger effects for handholding, suggesting that additional mechanisms are at play [9,63].

## Intense situations

In Studies 1–3 the preference for handholding (measured either by direct choice or by comparing ratings) was even stronger in intense situations. As detailed in the introduction, handholding allows for better two-way sensory communication than stroking, which could make it more effective in homeostatic emotion regulation. Intense situations might be more likely to cause an abrupt shift in internal states, requiring a quick return to the organism's baseline state–i.e., preservation of homeostasis. In contrast, non-intense situations might require a shift in the long-term biological setpoint, rather than a return to the norm [5]. Stroking may be especially suited for this role (sometimes termed allostatic regulation [5,64]). For example, in safe and relaxed situations in the presence of one's partner, stroking may help move the long-term biological setpoint, allowing the individual to be less vigilant in such situations in the future, freeing resources for other tasks. Indeed, gentle stroking has been shown to be prevalent among romantic partners [65] and to play a role in emotional communication of love and affection [66]. Future studies could compare situations like the ones presented in this study to situations which involve even less specific emotions than the low intensity situations used in the current study (e.g., a nondescript evening at home with one's partner). In such situations, stroking might even be preferred over handholding.

Notably, this effect was not found in Study 4. We attribute this to low power, or alternatively to a ceiling effect. The situation labeled as not intense—during childbirth, between contractions–might have been considered intense relative to everyday situations. Indeed, the main touch type effect in this study was the largest effect size of all four studies, supporting this assumption.

### Exploratory findings

**Valence and physicality.**   Exploratory findings for valence and physicality are summarized in S7 Table in the S1 Appendix. Focusing on moderation of touch type preference by valence and physicality, most effects were only significant in a single part of a single study, including three-way interactions and moderation of handholding preference by valence. Thus, the results support the notion that the effects are not limited to positive or negative-valenced situations. While our discussion is somewhat focused on regulation of negative emotions, which are the vast majority of emotion regulation instances reported in everyday life [67], our findings demonstrate that handholding is equally preferred to regulate either type of emotion.

The only effect which was somewhat consistent was the moderation of handholding preference by physicality–in Studies 1 and 2 (except for when using dichotomous measures in the second part of Study 2), the preference for handholding over stroking, as reflected in higher ratings, was even stronger in emotional situations than in physical ones. Simple slope analyses revealed that while ratings for stroking were similar between physical and emotional situations, ratings for handholding were higher in emotional situations, defined here as situations which did not directly involve the body (e.g., winning or losing money) as opposed to physical situations, defined here as situations which directly involved the body (e.g., falling down the stairs, dancing). A possible explanation could be that stroking involves both top-down processes (which may be more prominent in emotional situations) and bottom-up processes (which may be more prominent in physical situations), leading its effects to be comparable across these types of situations; handholding, on the other hand, seems to involve mostly top-down processes, which may explain why it is especially preferred in emotional situations. Still, the overall preference for handholding was still significant in physical situations, indicating that despite this moderation effect the main effects described above exist in both physical and emotional situations. Importantly, as this moderation effect was only an exploratory analysis, it should be treated as provisional until replicated in future studies.

**Blood/Injection phobia.**   Contrary to our expectations, the preference for handholding was weaker for participants with especially high blood/injection phobia. Simple slope analyses revealed that ratings for handholding were similar between participants with varied levels of phobia, while ratings for stroking were higher for participants with high levels of phobia. This effect might stem out of a ceiling effect for ratings of handholding by participants with high levels of phobia. Handholding was rated higher in general in this study than in the first two studies; Importantly, although the difference was not significant, participants with high levels of phobia rated all types of touch higher than participants with low levels of phobia. Thus, it is possible that participants with high levels of phobia rated handholding close to the maximal ratings possible, leaving less room for them to increase those ratings when compared to ratings of stroking. Still, the overall preference for handholding was still significant for participants with high levels of phobia, indicating that despite this moderation effect the main effects described above still exist for people with varied levels of injection phobia. Importantly, as this moderation effect was only an exploratory analysis, it should be treated as provisional until replicated in future studies.

**Cultural differences.**   No moderation of touch preference by culture was found. While this suggests that the differences in touch preference are at least somewhat robust across cultures, the sample size for Study 4 was relatively small, suggesting that this might simply be a null result due to low power.

## Implications for future research

The findings of the current study have several implications for future research on emotional regulation via touch. First, as participants have shown a clear preference for handholding, at least in hypothetical and recalled situations, research on touch as a form of emotion regulation should strive to include handholding conditions. Importantly, these studies should also attempt to compare the effectiveness of different types of touch in different top-down contexts (e.g., when trying to regulate intense vs. non-intense emotions). Effectiveness studies could also expand to include other types of touch, such as hugging.

On the other hand, while the current study examined a variety of situations, it could not explore every possible one. More subjective preference research should be conducted to investigate whether there are situations–involving emotion regulation or otherwise–in which people prefer stroking over handholding. Finally, if future effectiveness studies find that stroking is more effective than handholding as a form of emotion regulation, interventions should be developed to encourage people to use stroking in their everyday life.

## Limitations

While the four studies discussed in the current paper cover the question of subjective touch preference from a variety of angles, there are still several limitations.

First, while the study included both hypothesized and recalled situations, it did not include live ones. Future studies could put participants in actual situations requiring emotion regulation or use experience sampling methods to track participants as such situations naturally occur. This would allow them to test whether preferences in the moment are similar to hypothesized and recalled ones. Relatedly, in the second part of Study 4 some of the participants received the types of touch they were rating during their actual childbirth experience, while some did not, and were rating them hypothetically, a difference which may have affected the results. As only 7 women (under 20%) received stroking and only 11 women (less than 35%) did not receive handholding, analyzing these subgroups would be underpowered. That said, Studies 1–3 conclusively show that even when all situations are hypothetical, handholding is preferred over stroking.

Second, in Studies 1 and 2 situations were classified as physical or emotional (non-physical). We did not ask each participant to code situations on this axis, and as such cannot verify the validity of this classification. However, the physicality classification has no bearing on our main hypotheses concerning touch preference and intensity.

Third, as detailed above, the study used videos to present optimal stroking without biasing participants with an overly detailed description of stroking compared to handholding. Of course, while the videos were carefully designed to reduce possible biases–by using neutral backgrounds, not showing faces, etc.—some characteristic of the videos which was not controlled for might still have affected results. Some possible factors include video quality (e.g., non-steady camera, shadows), the fact that the actors were all in opposite-sex relationships while participants could have been of any sexual orientation (and in any type of relationship), or the fact that both types of touch were depicted in the air (i.e., not sitting down, or close to the body). This is somewhat mitigated by the first part of Study 4, in which types of touch were

presented verbally, without videos, and still "stroking" was overwhelmingly less common as a form of support for women giving birth than "handholding".

Finally, most participants were from Western, high-income countries, limiting the generalizability of the findings. Study 4 partially addressed this concern by including both Hebrew and Arabic-speaking participants; however, it should be noted that the sample size was small and the study was run in an Israeli (i.e., Western and high-income) context.

## Conclusion

In conclusion, the results of four studies demonstrate a consistent subjective preference for handholding over stroking as a form of emotional regulation, especially in intense situations. We suggest several possible explanations for this phenomenon, including the cultural familiarity of handholding, top-down processes having stronger effects than bottom-up processes on subjective preference, handholding being a more passive form of touch, and handholding's unique capability to establish two-way sensory communication, creating an optimal feedback loop.

## Supporting information

**S1 Appendix. Additional analyses.**
(DOCX)

## Author Contributions

**Conceptualization:** Haran Sened, Chen Levin, Manar Shehab, Naomi Eisenberger, Simone Shamay-Tsoory.

**Formal analysis:** Haran Sened.

**Funding acquisition:** Naomi Eisenberger, Simone Shamay-Tsoory.

**Investigation:** Haran Sened, Chen Levin, Manar Shehab.

**Methodology:** Haran Sened, Chen Levin, Manar Shehab, Simone Shamay-Tsoory.

**Project administration:** Haran Sened, Simone Shamay-Tsoory.

**Resources:** Simone Shamay-Tsoory.

**Supervision:** Naomi Eisenberger, Simone Shamay-Tsoory.

**Visualization:** Haran Sened.

**Writing – original draft:** Haran Sened.

**Writing – review & editing:** Haran Sened, Naomi Eisenberger, Simone Shamay-Tsoory.

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
