## [Decision Letter · Decision Letter 0]

15 Aug 2022

PONE-D-22-07411I Wanna Hold Your Hand:

Handholding is Preferred over Gentle Stroking for Acute Emotion RegulationPLOS ONE

Dear Dr. Sened,

Thank you for submitting your manuscript to PLOS ONE. After careful consideration, we feel that it has merit but does not fully meet PLOS ONE’s publication criteria as it currently stands. Therefore, we invite you to submit a revised version of the manuscript that addresses the points raised during the review process. Please pay particular attention to addressing the reviewer's comments regarding the framing of your Introduction and Discussion, and their recommendations regarding the presentation of the methods and results.

Please note that we have only been able to secure a single reviewer to assess your manuscript. We are issuing a decision on your manuscript at this point to prevent further delays in the evaluation of your manuscript. Please be aware that the editor who handles your revised manuscript might find it necessary to invite additional reviewers to assess this work once the revised manuscript is submitted. However, we will aim to proceed on the basis of this single review if possible.

We look forward to receiving your revised manuscript.

Kind regards,

Jamie Males

Editorial Office

PLOS ONE

https://journals.plos.org/plosone/s/file?id=ba62/PLOSOne_formatting_sample_title_authors_affiliations.pdf".

“The work was funded by a MINDSS grant awarded to the first author by the University of Haifa and by a U.S. - Israel Binational Science Fund (BSF) grant awarded to the fourth and fifth authors.”

4. Thank you for stating the following in the Funding Section of your manuscript:

“The first author’s work on the project was funded by a MINDSS grant from the University of Haifa faculty of social sciences. The fourth and fifth author’s work on the project was funded by a grant from the Bi-national Science Fund (BSF).”

“The work was funded by a MINDSS grant awarded to the first author by the University of Haifa and by a U.S. - Israel Binational Science Fund (BSF) grant awarded to the fourth and fifth authors.”

6. We note that you have referenced (Spielberger CD, Jacobs G, Crane R, Russell S, Westberry L, Barker L, et al. Preliminary manual for the state-trait personality inventory (STPI). Unpublished manuscript, University of South Florida, Tampa. 1979; which has currently not yet been accepted for publication. Please remove this from your References and amend this to state in the body of your manuscript:  [Unpublished]”) as detailed online in our guide for authors

http://journals.plos.org/plosone/s/submission-guidelines#loc-reference-style.

Reviewers' comments:

Reviewer's Responses to Questions

**Comments to the Author**

1. Is the manuscript technically sound, and do the data support the conclusions?

Reviewer #1: Yes

2. Has the statistical analysis been performed appropriately and rigorously? 

Reviewer #1: No

3. Have the authors made all data underlying the findings in their manuscript fully available?

Reviewer #1: Yes

4. Is the manuscript presented in an intelligible fashion and written in standard English?

Reviewer #1: Yes

5. Review Comments to the Author

Reviewer #1: The authors present a series of online studies where handholding is compared to arm stroking, in different intensity situations. They make the excellent distinction that handholding is a truly two-way (active/giving and receiving touch) interaction, whereas stroking is much more one-way (passive/receiving touch). The authors could make even more of this point throughout, where touch becomes a stronger form of communication in handholding. I congratulate the authors on the open way their studies are presented, yet the organization of the results is confusing. Overall, I am not convinced about the statistical analyses used. They seem incomplete, yet repetitive, and do not seem to correct for multiple tests, as well as using unjustified one-way t-tests. Also, the grounding of the work in previous literature needs to be revisited. However, the manuscript is novel and opens up new ideas into social touch mechanisms, adding new insights to the literature.

Introduction

- I do not think the study is well-grounded in the actual findings on affective touch. Many studies investigate the pleasantness of stroking touch and a few have conducted microneurography and shown that C-tactile fibers fire well to slow, gentle touch. However, C-tactile afferents are just one part of affective touch. The authors say, ‘Gentle stroking entails slowly stroking skin regions with C-tactile afferents…’ but this is imprecise, as gentle stroking will readily activate all types of mechanoreceptive afferent. Although CTs fire well to gentle skin stroking, it is clear that affective touch is not strictly linked to CT firing, where the relationship breaks down when other factors are added. Cognitive factors can influence gentle stroking perception, e.g. changes in pleasantness with different odors (Croy et al, 2014 Plos One). Such a breakdown in the CT vs. affective touch relationship has also been demonstrated, when touch was delivered at different temperatures and CT firing no longer correlated with pleasantness (Ackerley et al, 2014 J Neurosci). Further, a number of studies have shown little differences in the perception of pleasant touch on hairy arm and glabrous hand skin (for an overview, see Cruciani et al, 2021 Neurosci Biobehav Rev). Thus, affective touch is more complicated than simply CT afferent firing.

- In the next sentence, they say ‘…activates regions in the posterior insular cortex and the mid-anterior orbitofrontal cortex that are not activated by other types of touch’. Again, this is imprecise, as it is well-known that posterior insula is activated by all types of touch and the meta-analysis by Morrison (2016, Hum Brain Map) found that in general, posterior insula is more likely to be activated for affective touch, S1 for discriminative touch, whereas S2 is well-engaged in both; however, in no way were areas fully selective for one type of touch or the other.

- The authors say ‘no significant differences were found between handholding and gentle stroking, although effects of handholding were greater in every case’ for the work by Reddan et al (2020); however, if there were no significant differences in this work, they cannot claim that the effects of handholding were greater.

- The paper by Schirmer et al (2021, Emotion) on different types of touch may be useful to add to the introduction, in terms of the topographies of touch with others and the social implications.

Methods

- Would it be possible to add age details (e.g. range) for each study? I am also presuming that the sample for Study 4 was only females, but it would be good to state this too.

- It would be good to have the ethical section at the beginning of the methods, as normally done.

Results

- Please can the number of decimal places be reduced in the tables? Only one (maybe two) decimal places are relevant for the data, too many numbers make the table difficult to interpret and I do not think such accuracy is relevant for these ratings (also for the t values).

- I do not understand why the data have been analyzed twice. Two mixed lineal models were conducted with intensity as a dichotomous variable and as a continuous variable, but there is no clear reason for this and it confuses the results.

- On p.22, I do not understand why the authors say (two times) that ‘this was only true in a one-tailed significance test’ and have p values of >0.05. I do not think they are justified in using a one-tailed t-test (why not corrected, post-hoc tests) and these p values are not significant. Also, just underneath, a p value of 0.094 is quoted as significant, when it is not.

- Top of p.23, why are the results for non-intense situations not stated?

Discussion

- Although well-written, the first part of the discussion goes too far. All the studies were online and featured videos and hypothetical situations. Therefore, the authors need to be clear that these were not actual, real, in-person situations, where the results could be different.

- Near the bottom of p. 24, the authors say, ‘…as opposed to stroking, which relies mainly on bottom-up processes’. I do not fully agree that stroking specifically relies mainly on bottom-up processes. Although receiving stroking is passive touch, the situation will determine the perception, including various top-down effects. Such top-down effects may be more relevant for active touch in hand-holding, but they are present in both situations.

- In the middle of p. 25, the authors say, ‘even if handholding is found in the future to be less

effective than stroking’, but what does this mean? Less effective in what way? I am not sure if I follow all the section on ‘effectiveness’, as this concept has not been well-defined throughout. Do you mean how effective touch is to console/comfort someone?

Minor comments

- p. 14, ‘hypothesize themselves in different hypothetical situations’ could be revised for clarity. Imagine themselves?

6. PLOS authors have the option to publish the peer review history of their article (what does this mean?). If published, this will include your full peer review and any attached files.

Reviewer #1: No

---

## [Author Response · Author response to Decision Letter 0]

8 Sep 2022

Responses are included in the attached response file.

---

## [Decision Letter · Decision Letter 1]

25 Oct 2022

PONE-D-22-07411R1I Wanna Hold Your Hand:

Handholding is Preferred over Gentle Stroking for Acute Emotion RegulationPLOS ONE

Dear Dr. Sened,

Thank you for submitting your manuscript to PLOS ONE. After careful consideration, we feel that it has merit but does not fully meet PLOS ONE’s publication criteria as it currently stands. Therefore, we invite you to submit a revised version of the manuscript that addresses the points raised during the review process.

We look forward to receiving your revised manuscript.

Kind regards,

Rochelle Ackerley

Guest Editor

PLOS ONE

Additional Editor Comments:

Thank you for revising your manuscript. I am sorry that the review process has been long for this submission. To explain the situation, Plos One had difficulties finding an Editor. I was the first reviewer in the initial round and I am now the Guest Editor. I sent your manuscript out for review again and three experts have commented on it. I am no longer a reviewer, but I would like you to look over the new reviewers' comments and answer them to your best ability. Your work is very interesting and if you take the comments on board, this will improve your manuscript. It is especially important to consider the points about the analysis and including previous literature. I look forward to seeing another version of the manuscript.

Reviewers' comments:

Reviewer's Responses to Questions

**Comments to the Author**

1. If the authors have adequately addressed your comments raised in a previous round of review and you feel that this manuscript is now acceptable for publication, you may indicate that here to bypass the “Comments to the Author” section, enter your conflict of interest statement in the “Confidential to Editor” section, and submit your "Accept" recommendation.

Reviewer #2: (No Response)

Reviewer #3: (No Response)

Reviewer #4: (No Response)

2. Is the manuscript technically sound, and do the data support the conclusions?

Reviewer #2: No

Reviewer #3: Yes

Reviewer #4: Partly

3. Has the statistical analysis been performed appropriately and rigorously? 

Reviewer #2: No

Reviewer #3: Yes

Reviewer #4: I Don't Know

4. Have the authors made all data underlying the findings in their manuscript fully available?

Reviewer #2: Yes

Reviewer #3: Yes

Reviewer #4: Yes

5. Is the manuscript presented in an intelligible fashion and written in standard English?

Reviewer #2: Yes

Reviewer #3: Yes

Reviewer #4: Yes

6. Review Comments to the Author

Reviewer #2: This is a very interesting and potentially important study. However, there are too many open issues concerning both the study methods and analyses that make it impossible to clearly evaluate the actual value of the data. I list below the things that require further explanation and re-analysis.

Page 3: The idea of allostasis was already implemented in early models of homeostasis. Now, the term allostasis is typically used to refer to deviations that have some sort of clinical implication or are disease relevant. Apart from this issue, I don’t quite see the usefulness of introducing the concept of allostasis as it is not needed for the paper. Indeed all that’s needed is the idea of a feedback loop.

Page 4: In the last paragraph, the critical distinction is that handholding enables reciprocal touch whereas stroking does not. So the classical distinction between reciprocal and non-reciprocal touch really is central here and should be introduced accordingly.

Page 8: There is no value in doing post-hoc power analyses. This should be deleted.

Page 9: I found no touch videos following the link. I only found a zip archive with photos and it wasn’t clear to me how they were being presented. Their names contained numbers and letters that were little intuitive. As this is the key manipulation, what exactly was shown needs to be better explained and illustrated in the main manuscript. Just from looking at Figure 1 it would seem that there was a lot more skin contact for the hand holding condition than for the stroking condition and that would constitute a confound. I believe this was addressed in Study 2 but this isn’t explained or visually supported in any way.

More information about the situations is needed. Please plot means and CIs for ratings for each of the selected situations. Report the rating scale endpoints. As suggested below, it would be useful to do this for each study separately using the study participant data rather than the data obtained in the pre-experimental rating.

Page 11: Please report the software/tool you used to do your power analysis. This is not trivial in the context of mixed modeling.

Page 13: Its a bit disconcerting that intensity flip flops between different samples. This suggests that it is not a stable measure and likely not useful to test your hypotheses. In your results you report analyses using the participant’s own ratings as a variable. Given these issues that would be the more reasonable way to go. So I would do away with the pre-experimental ratings and trying to categorize situations into intense and not intense. Your stats is a bit dense as it is. No need to add to this by including analyses that aren’t informative… Note that this also raises concerns about how reliable/consistent your valence and physicality ratings were. Why would we trust those? Again, I would suggest dropping those. As a reader I am not convinced…

The reader is variously referred to S1 appendix and supplementary materials. Are they the same? I could only find S1 appendix with this submission. But some of the information (e.g., situation intensity ratings for the different studies) was not provided.

Page 15: Here we learn for the first time that there are two different stroking conditions. This is a bit confusing. I also don’t understand how the videos were assigned to situations. What exactly were the “video sets”? Again this is the key manipulation and needs to be fully explained.

The description of the task is a bit misleading. What you say in the text and what is shown in Figure 1 are at odds. If I understand Figure 1 correctly, then participants did not select what type of touch condition they preferred. Instead, they rated for each touch condition, how it would make them feel in a given situation.

Page 16: A 5-point Likert scale is not a continuous measure. It is an ordinal measure and should be treated as such. I recommend the ordinal package in R.

Please report the software and packages you used to conduct your statistical analysis.

How did you convert what you call “preference rating” (which is a misnomer) into numbers? This is necessary to understand the data in Table 1. Later sections of that table are non-legible due to line breaks. There are variables in there that haven’t been introduced (e.g., UCLA). The table really isn’t helpful. I suggest presenting key information in figures, and place a cleaned up version of the table in your supplementary materials. Current figure 2 just shows the models not the data. Please find a proper way to plot your data. I suggest rain-cloud plots for their transparency (https://www.r-bloggers.com/2021/07/ggdist-make-a-raincloud-plot-to-visualize-distribution-in-ggplot2/).

So I assume you had 8 values for each cell per participant on average? If you don’t model random slopes, you are increasing your type 1 error. If you enter more than one value per cell per participant you should add random slopes or include situation as a factor. Alternatively, if these models don’t converge, as you are not interested in situation you should use a mean across situations. Also please report missing cells as a function of subjects. I would assume there are some. One would hope there are not too many including cases that rate each situation equally intense…

Earlier in the manuscript you state that you tested both positive and negative situations and different types of what you call physicality. What happened with these manipulations in your analyses? This goes back to the issue that its not clear what exactly was presented in terms of combinations of the situations and touch types.

Please report how you determined effect sizes from your mixed models. This is a bit tricky for LMEs.

How did you follow-up interactions? Did you do this within the full model or did you run simpler separate models? Did you look at the intensity effect for each level of touch? Please spell things out for your reader.

Page 25: Now here you really do look at preference. Please be clear throughout your manuscript what your items actually measure.

I’m unclear what you did with the logistic regression. It would seem that you treated data points within participants as independent. That would be inappropriate. How can you have a significant touch effect when touch preference (handholding vs stroking) was your DV? Or what was your DV? More information is needed.

For study 4, did you specifically ask about hand holding and hand stroking? What if there was other stroking touch that didn’t target the hand? That would seem to be a problem with interpreting the frequency of handholding vs stroking.

The results section ends with reports on valence and physicality effects. As mentioned earlier I’m unclear how these variables intersect with what else was presented in each study. No information is provided about how they were statistically analyzed. Hence, results cannot be evaluated.

I am not reading the discussion as I don’t see much point. There simply are too many question marks concerning the paradigm and analyses. In my opinion this needs another major revision in which the methods and result sections are improved. The results should include only appropriate analyses that are methodologically justified and properly explained.

Reviewer #3: In this four-experiments study, authors aimed to show that handholding was a stronger social touch than stroking to regulate intense situations. This is an interesting and very relevant study – however I have several comments that could be taken into account in order to improve the manuscript.

Overall, it is not clear why the authors chose handholding and stroking over other types of social touch. Especially in the abstract stating that “social touch highlights two forms of touch: handholding and stroking”, is quite limiting (and not true – see for ex Hertenstein et al. 2009; McIntyre et al. 2022). Authors should rephrase and discuss limitations accordingly. And in the introduction, page 4 please back up your discourse by references when stating that there are two main forms of consoling touch.

Moreover, there seems to be a general bias towards the authors hypothesis of preference for handholding in general: indeed, by looking at the gif presented to the participants, the naturalness of the handholding is much stronger than for the stroking. Authors could try to control for this (asking new participants to rate naturalness) and/or add it as a strong limitation. On the same line, the stimuli picturing the stroking conditions are not very well filmed (camera moving) and it is rare to perform this type of touch on the move / in the air. Moreover, there are quite a lot of different types of stroking so authors should acknowledge this.

The study by Burgoon 1991 is interesting however indeed they did not test arm stroking, so authors should rephrase their conclusion/reformulate as arm touching is quite different to arm stroking (stroking hasn’t been shown to communicate dominance but more comfort). Also, slow vs fast stroking does not communicate the same emotions (see for ex Kirsch et al. 2018).

I would advise to refrain from talking about C-touch for slow caress like stroking, just use the slow gentle stroking or caress like stroking terminology.

In the introduction – before the overview – it would be nice to justify the reason why you are investigating “touch provision vs reception, situation valence, emotional vs physical and cultural differences”. Moreover, actually you are not testing cultural differences (or results missing).

Authors should justify from the introduction the reason why they are conducting the studies only in people being in a heterosexual romantic relationship. And then discuss the implied limitations.

In study 4 – authors state that ‘in cases where they were not provided a particular type of touch..’: how often does this happened? As it is very different to rate received vs hypothetical touch.

For the touch preference rating: this section is very unclear. Please clarify what the participants had to do: rate each video or choose the one they preferred and rate only this one? Probably the terminology used is confusing: as it is not a preference rating but how it would make people feel. Please reformulate.

For data analyses: why coding 0,5 vs -0,5 the type of touch, and not use a categorical variable? Moreover, could you have run simple ANOVA on your data? Please better justify you analyses.

It is unclear what is the difference between table p18 and table p20? Why including UCLA ratings there is you are not analyzing it? Moreover, it is unclear in these tables how you take into account valence (positive vs. negative scenario).

Table 2 is unclear: if it represents the results of the mixed linear model could you please add the b, CI and R2? Also as in the text you are mentioning slopes p24 but they are not reported.

Please add in the Statistical Analysis section all the other analyses you have run: how you analyzed the subjective preference, recalled perception etc.. As for example the chi-square analyses are unclear.

Figures in general should be improved (titles, axes, legends). For example, in figure 2 what is ‘improvement’? The same scales should be used in all the graphs to be comparable; and in the legend please indicate what are the scales - for example why is intensity represented from -2 to 2, when the participants answer on a 4-point Likert scale?

Discussion:

It is unclear how results support this statement “It is much more likely to be performed in real life situations involving intense emotions”. Please justify and clarify.

Please add all limitations mentioned in points above.

Reviewer #4: It was a pleasure to able to review this manuscript titled: I wanna hold your hand: handholding is preferred over gentle stroking for acute emotion regulation. I have read the manuscript with great interest as it (supposedly) addresses an open question in the field of social touch: what types of touch are more suited to provide emotional support. I found the study interesting in principle and well powered. I really appreciate the large sample size and the fact that all the studies were preregistered. However, I found the process of reading and understanding the paper very challenging. I provide more specific comments below, but generally my main concerns are regarding the introduction and the discussion that should be completely revised, in my opinion. Too many generalizations are made, working hypotheses are cited as evidence, and scientific rigor is lacking. Furthermore, the present studies are not grounded in the literature, the results are very difficult to follow, the discussion does not do what a discussion is meant to do, which is to discuss the results in the context of what was already known in the field. Overall, I have noticed a lack of clarity and precision throughout the manuscript, and the conclusions are not fully supported by the results. Therefore, I am not in the position to recommend acceptance of the manuscript in the present form. I am providing some suggestions that could potentially help to improve the manuscript in turn below:

• Abstract: It is not clear from the abstract that the studies were conducted online. This should absolutely be included in the abstract, otherwise it can be misleading.

• Page 5: On the second page of the introduction, it remains unclear whether “consoling touch” should be classified as a touch that targets homeostatic or allostatic regulation. I think it is important to clarify this, or to say that consoling touch can address both needs…?

• Page 5: When talking about the positive effects of handholding and touch more generally, I think it is important to stress the fact that such effects are modulated by individual differences in patterns of relating (e.g., attachment). Furthermore, there is a large body of studies that are very relevant and have just been neglected here. Please see below:

Krahé, C., Paloyelis, Y., Sambo, C. F., & Fotopoulou, A. (2014). I like it when my partner holds my hand: Development of the responses and attitudes to support during pain questionnaire (RASP). Frontiers in Psychology, 5, 1027.

von Mohr, M., Kirsch, L. P., & Fotopoulou, A. (2017). The soothing function of touch: affective touch reduces feelings of social exclusion. Scientific reports, 7(1), 1-9.

von Mohr, M., Krahé, C., Beck, B., & Fotopoulou, A. (2018). The social buffering of pain by affective touch: a laser-evoked potential study in romantic couples. Social cognitive and affective neuroscience, 13(11), 1121-1130.

Krahé, C., von Mohr, M., Gentsch, A., Guy, L., Vari, C., Nolte, T., & Fotopoulou, A. (2018). Sensitivity to CT-optimal, affective touch depends on adult attachment style. Scientific reports, 8(1), 1-10.

Krahé, C., Paloyelis, Y., Condon, H., Jenkinson, P. M., Williams, S. C., & Fotopoulou, A. (2015). Attachment style moderates partner presence effects on pain: a laser-evoked potentials study. Social cognitive and affective neuroscience, 10(8), 1030-1037.

Krahé, C., & Fotopoulou, A. (2018). Psychological and neurobiological processes in coping with pain: The role of social interactions. In The Routledge International Handbook of Psychobiology (pp. 73-92). Routledge.

Burleson, M. H., & Quigley, K. S. (2021). Social interoception and social allostasis through touch: legacy of the somatovisceral afference model of emotion. Social neuroscience, 16(1), 92-102.

• Page 6: “ …stroking is uniquely associated with elevated oxytocin levels (13)”. I invite the Authors to be more cautious about this. I would advise to edit this sentence as to clarify that this is a working hypothesis. (e.g., ….it has been suggested that …).

• Page 6: When saying that handolding and C-touch types of touch are effective forms of emotion regulation, I would like to think this is true and it is something that many labs have been trying to prove, but the authors must be more precise and should avoid mentioning this as a fact. Also, which types of emotions are they referring to? Citing some papers to support this statement would be helpful.

Updated after continuing reading: But then, when I continue reading, I find the paragraph saying that “existing studies in the field have focused on testing the effectiveness of touch as a method of emotion regulation, with most studies …”. So, the Authors now question something that was given as certain a few lines above? Perhaps this section should come first, or the Authors might want to be a bit more clear here.

• Page 6: “…as opposed to stroking, handholding allows both participants to sense each other equally and simultaneously using their hands, which are more effective as tactile sensory organs than the forearms”. I find sentences like this one quite problematic for the following reasons:

1. The sensory organ in touch is always the skin. The hands and the forearms are not sensory organs, they are body sites.

2. Stroking can also refer to palm against palms, and as such, the proposed difference between handling and stroking would not apply anymore.

3. Stroking can also be delivered from forearm to forearm, and as such both individuals would be equally and simultaneously touched

4. Touch is always a two-way sensory communication, so I find this statement very weak as a basis for the hypothesis of this study.

5. To me, one of the main differences between stroking and handholding (that is not mentioned at all) has more to do with the fact that handholding is static, while stroking is a dynamic type of touch.

• Page 6: “….studies compared general pleasantness of touch between touching arm and hand skin found little difference (18).” The Authors are citing the meta-analysis of Cruciani et al in an incorrect way, for what I understood. Cruciani and colleagues discussed differences between hairy and non-hairy skin. Here the Authors discuss differences between arm and hand. The hand has both hairy (dorsal part of the hand) and non-hairy skin (palms), so I think the Authors should double check and be consistent with the use of terms throughout the manuscript, otherwise it is all very confusing.

• Generally, in the introduction, the Authors do not mention various, very important pieces of knowledge in the field of social touch that can be very relevant to build the framework for this paper:

1. CTs can be found also on the non-hairy skin of the body (e.g., palms); this body part is involved in handholding: Watkins, R. H., Dione, M., Ackerley, R., Backlund Wasling, H., Wessberg, J., & Löken, L. S. (2021). Evidence for sparse C-tactile afferent innervation of glabrous human hand skin. Journal of Neurophysiology, 125(1), 232-237.

2. CTs have been associated also to deep pressure touch – which is supposedly is involved in handholding: Case, L. K., Liljencrantz, J., McCall, M. V., Bradson, M., Necaise, A., Tubbs, J., ... & Bushnell, M. C. (2021). Pleasant deep pressure: expanding the social touch hypothesis. Neuroscience, 464, 3-11.

3. There are studies already showing individual preferences for the types of touch (see for example, Perini, I., Olausson, H., & Morrison, I. (2015). Seeking pleasant touch: neural correlates of behavioral preferences for skin stroking. Frontiers in behavioral neuroscience, 9, 8.)

• Page 6: When citing the work of Reddan and colleagues, I think it is very important and informative to also cite the (several) publications from Charlotte Krahé and Aikaterini Fotopoulou that have investigated the role of touch in modulating pain (I have mentioned them in one of my points above).

• Page 7: “While research on the effectiveness of these types of touch has achieved mixed results…” What are these mixed results the Authors are referring too? I do not thing that have been discussed properly. What are the controversies/open questions/outstanding issues in this field? This is unclear and should be discussed here.

• Page 8: I think this paper would be very relevant here, and I am very surprised it has not been mentioned at all:

von Mohr, M., Kirsch, L. P., & Fotopoulou, A. (2017). The soothing function of touch: affective touch reduces feelings of social exclusion. Scientific reports, 7(1), 1-9.

• Page 8: “We did not have any hypothesis concerning valence and physicality”. It is very hard to understand what the Authors mean with valence. I suggest including some examples in parathesis to explain what you mean by “valence”. Furthermore, I am very surprised that no hypothesis was formulated about valence. Surely one would expect that a positive or a negative emotion would make a difference (if I understand what the Authors mean by valence, as it has not been spelled out)…

• Page 8: “…we hypothesized that the preference for handholding would be stronger in intense situations, involving strong, acute emotions”. Again, this is very unclear. What do you mean by “strong, acute emotions”?

• Page 9: Why the Authors did choose Arabic-speaking and Hebrew-speaking participants to investigate cultural differences? This implies some differences in emotion regulation or in the use of touch due to the spoken language, but this is what I imagine, as it has not been explained, contextualized, rationalized here.

• Page 10, Open data and preregistration: I must say that this is the first time I see the word “videos” so far and it came a bit of nowhere. I think that the fact that the experiments are done online, and videos are used should be introduced much earlier in the manuscript (e.g., introduction). Readers go through the hypotheses without knowing a very important detail about the used approach. There are several studies that used videos of touch, and this opens the door to a field of studies (sometime referred to as vicarious touch) that has not been mentioned at all in the introduction and I think it is of crucial importance here. Participants are not simply imagining of being touched (as I originally thought) but they are watching videos of people touching. I strongly encourage the Authors to consider adding some references from the vicarious touch relevant and introduce this topic in the introduction.

• Page 17, Statistical analysis: Which software/package was used for data analysis? This information should be added.

• Page 25-18, Results: I must say that I found extremely difficult to follow and understand the results. The results are presented in tables with no sufficient explanation, and I think that the main results should also be reported in the text. Furthermore, expressions such as “almost always” “in almost all studies” should be avoided in the results section. I would like to see more precision and clarity. I am wondering if reporting the results study by study rather than organized in topics would be more helpful.

• Page 30, Discussion: I think that it is quite confusing and beyond the point to answer the question “Why is handholding preferred over stroking?” by referring to Greek literature and the story of Adam and Eve. The Authors should stay closer to the data and what it is known in the scientific literature in humans. The discussion overall is too brief and does not help in understanding the results.

• Page 30-31, it is unclear to me why top-down factors play a more important role in handholding as compared to stroking. There is a huge increase in research showing the impact of intentionality, cognitive factors, and top-down factors in stroking studies, and these would go completely against this explanation. See for example:

Sailer, U., & Leknes, S. (2022). Meaning makes touch affective. Current Opinion in Behavioral Sciences, 44, 101099.

Sailer, U., Hausmann, M., & Croy, I. (2020). Pleasantness only? How sensory and affective attributes describe touch targeting C-tactile fibers. Experimental Psychology, 67(4), 224.

Ellingsen, D. M., Leknes, S., Løseth, G., Wessberg, J., & Olausson, H. (2016). The neurobiology shaping affective touch: expectation, motivation, and meaning in the multisensory context. Frontiers in psychology, 6, 1986.

McCabe, C., Rolls, E. T., Bilderbeck, A., & McGlone, F. (2008). Cognitive influences on the affective representation of touch and the sight of touch in the human brain. Social cognitive and affective neuroscience, 3(2), 97-108.

Furthermore, it is also unclear why the paper of Cascio et al that is cited here would support this view.

7. PLOS authors have the option to publish the peer review history of their article (what does this mean?). If published, this will include your full peer review and any attached files.

Reviewer #2: No

Reviewer #3: No

Reviewer #4: No

---

## [Author Response · Author response to Decision Letter 1]

24 Jan 2023

Reviewer responses are included in the attached file

---

## [Decision Letter · Decision Letter 2]

17 Feb 2023

PONE-D-22-07411R2I Wanna Hold Your Hand:

Handholding is Preferred over Gentle Stroking for Emotion RegulationPLOS ONE

Dear Dr. Sened,

Thank you for submitting your manuscript to PLOS ONE. After careful consideration, we feel that it has merit but does not fully meet PLOS ONE’s publication criteria as it currently stands. Therefore, we invite you to submit a revised version of the manuscript that addresses the points raised during the review process.

We look forward to receiving your revised manuscript.

Kind regards,

Rochelle Ackerley

Guest Editor

PLOS ONE

Additional Editor Comments:

Thank you for your revised manuscript and your patience in the review process. The expert reviewers do see the interest and significance of your work, where two suggest minor/no changes, whereas one suggests more in-depth revision. The review comments are important to take into account and I would appreciate it if the authors could address all the comments posed.

Reviewers' comments:

Reviewer's Responses to Questions

**Comments to the Author**

1. If the authors have adequately addressed your comments raised in a previous round of review and you feel that this manuscript is now acceptable for publication, you may indicate that here to bypass the “Comments to the Author” section, enter your conflict of interest statement in the “Confidential to Editor” section, and submit your "Accept" recommendation.

Reviewer #2: (No Response)

Reviewer #3: All comments have been addressed

Reviewer #4: All comments have been addressed

2. Is the manuscript technically sound, and do the data support the conclusions?

Reviewer #2: No

Reviewer #3: Yes

Reviewer #4: Yes

3. Has the statistical analysis been performed appropriately and rigorously? 

Reviewer #2: No

Reviewer #3: Yes

Reviewer #4: Yes

4. Have the authors made all data underlying the findings in their manuscript fully available?

Reviewer #2: No

Reviewer #3: Yes

Reviewer #4: Yes

5. Is the manuscript presented in an intelligible fashion and written in standard English?

Reviewer #2: Yes

Reviewer #3: Yes

Reviewer #4: Yes

6. Review Comments to the Author

Reviewer #2: Although the authors have made a significant effort to revise their manuscript, important issues have not been resolved. Moreover, I am concerned that their statements are too broad and unjustified by the data. They send the wrong message and may do more harm than good for the field. Below I explain my main theoretical/conceptual issues and then elaborate on some of the outstanding methodological/analytical problems.

I previously asked that the authors introduce the distinction between reciprocal and non-reciprocal touch in their introduction when discussing differences between handholding and stroking and the bidirectionality of the former, which is lacking for the latter. They wrote in their response that there is no research about this. However, this distinction has been around for decades. I suggest looking at the book by Hall and Knapp (https://www.amazon.com/Nonverbal-Communication-Human-Interaction-Knapp/dp/1133311598). Again, I consider this distinction central to the comparison they make and something that should be raised when introducing their research question. Ultimately, their handholding condition is active, whereas their stroking condition is passive and it is more than simply tactile processes that differ here. There are additional motor and proprioceptive processes that contribute to hand-holding. There is also the issue that simply holding onto something (be it another person or another object) could help regulate discomfort or pain.

I was now able to view the gif files. One thing I noted is that stroking seemed significantly faster than 3 cm/s. Perhaps time changed in the process of creating the gifs? Can the authors measure the speed in the images and document that?

In general, the stroking looks a bit mechanical rather than affectionate. Moreover, I don’t think the position of the people stroking is typical for the manner in which people stroke each other. When would you stroke someone’s forearm standing side by side? It is uncomfortable. This position is much more natural for handholding. For several of the clips the action is not in center and moves out of the frame. There are also significant shadows of the body parts on the walls that could affect perception.

I wonder whether body part matters. Could a rub on the back be more comfortable/preferred than rubbing of the forearm? In any case, I think that a generalization to all kinds of stroking is not possible here. The authors don’t compare hand-holding with stroking, they compare hand-holding with forearm stroking (from an unnatural position with a certain speed).

I had asked previously that more information about the situations in Study 1 and 2 be provided. I have not found this in the revision. I’m wondering what kind of emotions were examined (e.g., sadness, anger, fear)? Could stroking be more effective for low-arousal states such as dejection and handholding for high-arousal states such as fear? Indeed, I’m skeptical that the data allow the authors to conclude that emotion regulation in general benefits more from handholding than stroking. Emotion regulation is a very broad class of processes that includes the enhancement, maintenance and reduction of any kind of emotion.

Relatedly, it seems from the discussion that some of the situations were positive? Did participants opt for handholding to increase/maintain positive and/or reduce negative emotions? Much of the paper is written in a way that suggests touch is used to down-regulate negative feelings but it seems the data may not clearly speak to that and may be more nuanced.

So with all that’s said above, it is not appropriate to say that handholding is more effective than stroking for emotion regulation. Instead, the authors should carefully reflect on their methods/results and offer a more nuanced discussion and summary of findings in their title, abstract and main body of the paper.

Other concerns:

Page 14: „Continuous Situation Intensity and Valence“ I had mentioned in my previous review that these are not continuous measures.

I’m unclear about what the authors mean by intensity? How is it different from being very positive or negative? How were participants instructed to do the rating?

The logistic regression should include random intercepts and slopes. I’m also confused about the choice of test here. It would seem a chi-square test would be more appropriate. This is indeed what is used in Study 4.

I had asked previously that the rating analyses be revised in line with the statistical properties of the data. I don’t understand why the revised analyses are in the supplementary materials rather than the main body of the manuscript? Also, the analyses there are inappropriate. To analyze the ratings, the authors should run an ordinal model with all random slopes and intercepts, rather than an ordinal model with only random intercepts and a linear model with random intercepts and slopes, which is what they have now placed in the supplementary materials. Tables S10 and 11 show no degrees of freedom. They are also missing in the main text.

Follow-up tests of the interaction between intensity and touch were not reported. The authors say what package they used, but do not explain what they actually did. Did they use the simple_contrast function on their LME model? For the results, the reader is sent to the Appendix.

The figures have not improved. The authors count data points and participants without discriminating between the two. So they don’t present subject means and do not allow readers to get a sense of the variance in the data. There are a couple of rain cloud plots, but these are not fully presented. It seems the distributions were very broad and are cut off at the margin of the figure.

Page 30: „physical or emotional – on a 1-100 scale, with no verbal indicators for

endpoint“ How can the authors be certain that all participants used this scale as intended by them? I could consider ratings as reflecting the amount of pain that is being reduced or the amount that pain is being reduced to.

Page 32: „was strongly preferred behaviorally by people assisting women giving birth“ This was not examined. You simply examined the kind of touch women giving birth recall having received from their partner.

I looked at the OSF archives. I could only find some MTURK pilot data that was the same for Study 1 and 2. Otherwise I found no data.

Reviewer #3: Authors have addressed all my comments.

However, I have some comments on the revised version:

- Please make sure you proofread the paper as some grammatical errors are present (e.g. "was significantly differed")

- Some comments seem to not be included in the main text; while mentioned in the response to reviewers, please justify / change it. For example for experiment 1, authors states that they elaborated the chi-square results "beyond asking participants to rate ..." but I cannot find this paragraph in the revised version of the manuscript.

- On that note, the second part of this added paragraph is not totally clear to me. "Touch type chosen and intensity were not statistically independent (χ2(2)= 43.317, p < .001), meaning that handholding was preferred in intense situations." Could you please reformulate/correct ?

- Figure legends: could you expand the legends, especially for Fig1 and 2, explaining what are the images picturing? For fig 2: are these the 4 types of touch participants rate in all studies ? Add this info in the figure legend. For fig 1, is the first image picturing stroking or no touch ?

For figure 6 to 9 what does the legend "touch rating" represent ? please explain in the legend and on the figure as "touch rating" is not specific enough. Please be clearer on what is presented, as for now figures are difficult to understand.

- Study 2: why not include the chi-square analyses results in the descriptive part ?

- Page 31: touch reception ratings in study 4: which figure are you referring to? Probably figure 9. On that note, why representing the results differently here from the other studies ?

- Overall, I would advise to go again over the results section of each study to improve clarity.

Reviewer #4: The Authors have addressed my concerns to a satisfactory extent, and I believe that the manuscript is now suitable for publication.

7. PLOS authors have the option to publish the peer review history of their article (what does this mean?). If published, this will include your full peer review and any attached files.

Reviewer #2: No

Reviewer #3: No

Reviewer #4: No

---

## [Author Response · Author response to Decision Letter 2]

15 Mar 2023

We have responded to reviewer comments in the attached response file.

---

## [Editor Report · Decision Letter 3]

27 Mar 2023

I Wanna Hold Your Hand:

Handholding is Preferred over Gentle Stroking for Emotion Regulation

PONE-D-22-07411R3

Dear Dr. Sened,

We’re pleased to inform you that your manuscript has been judged scientifically suitable for publication and will be formally accepted for publication once it meets all outstanding technical requirements.

Kind regards,

Rochelle Ackerley

Guest Editor

PLOS ONE

Additional Editor Comments (optional):

Thank you again for your revision and for your patience. I understand that this has been a long time in revision, but I really do appreciate how well you have responded to the reviewers and dealt with this process. I am very pleased to accept this very interesting paper and congratulate you on this in-depth work.
---

## [Editor Report · Acceptance letter]

29 Mar 2023

PONE-D-22-07411R3 

I Wanna Hold Your Hand:
Handholding is Preferred over Gentle Stroking for Emotion Regulation 

Dear Dr. Sened:

I'm pleased to inform you that your manuscript has been deemed suitable for publication in PLOS ONE. Congratulations! Your manuscript is now with our production department. 

Kind regards, 

on behalf of

Dr. Rochelle Ackerley 

Guest Editor

PLOS ONE